# Non-equilibrium Sachdev-Ye-Kitaev model
# with quadratic perturbation

**Aleksey V. Lunkin**[1,2,3⋆] **and Mikhail V. Feigel'man**[2,4]

**1** Skolkovo Institute of Science and Technology, 143026 Skolkovo, Russia
**2** L. D. Landau Institute for Theoretical Physics, Kosygin Str. 2, Moscow 119334, Russia
**3** HSE University, Moscow, Russia
**4** Moscow Institute of Physics and Technology, Moscow 141700, Russia

⋆ alunkin@itp.ac.ru

## Abstract

We consider a non-equilibrium generalization of the mixed $SYK_4+SYK_2$ model and calculate the energy dissipation rate $W(\omega)$ that results due to periodic modulation of random quadratic matrix elements with a frequency $\omega$. We find that $W(\omega)$ possesses a peak at $\omega$ close to the polaron energy spliting $\omega_R$ found recently in [1], demonstrating physical significance of this energy scale. Next, we study the effect of energy pumping with a finite amplitude at the resonance frequency $\omega_R$ and calculate, in presence of this pumping, non-equilibrium dissipation rate due to low-frequency parameteric modulation. We found unusual phenomenon similar to "dry friction" in presence of pumping.


# 1   Introduction

The Sachdev-Ye-Kitaev model [2–5] modified by random quadratic terms in the Hamiltonian presents a valuable starting point to develop a theory of strongy correlated electron systems. Recent results [1] demonstrate an interesting interplay between soft-mode fluctuations (which dominates the infra-red behavior of the pure $SYK_4$ model) and $SYK_2$ terms of moderate magnitude. Namely, it was found that in presence of $SYK_2$ terms the soft-mode fluctuations become suppressed in a wide temperature range, and can be described by a kind of nearly Gaussian action. The crucial role in this picture is played by the bound-polaron solution obtained in [1] for some collective Bose field. The Liouville quantum mechanics approach [3,5] to the pure $SYK_4$ model can be recast into the form of the functional integral over the same Bose field, but in this case bound-states are absent and fluctuations are strong and non-Gaussian. We have shown [1] that quadratic perturbation lead to the formation of the polaron bound-state and thus to suppression of fluctuations at low energies.

In the present Letter we extend the approach of Ref. [1] into a broad field of non-equilibrium problems described by Keldysh functional integral methods. We study the simplest physical quantity that may shed some light on the physical significance of the polaron bound-state. Namely, we calculate the power dissipated in the system due to time-dependent modulation of the quadratic part of the Hamiltonian. In other words, we propose here generalization of the approach well-known [6–9] for non-interacting random Fermi-systems, where dissipation due to time-dependent periodic perturbation can be expressed in terms of parametric statistics of a Wigner-Dyson random-matrix ensemble.

## 2 The model

The Hamiltonian of the model has the form:

$$H = \frac{1}{4!} \sum_{ijkl} J_{ijkl} \chi_i \chi_j \chi_k \chi_l + \frac{i}{2} \sum_{ij} \Gamma_{ij}(1 + \Phi(t)) \chi_i \chi_j. \tag{1}$$

Here $J_{ijkl}$ and $\Gamma_{ij}$ are random Gaussian antisymmetric tensors. Their average values vanish, while $\overline{J_{ijkl}^2} = \frac{3!J^2}{N^3}$ and $\overline{\Gamma_{ij}^2} = \frac{\Gamma^2}{N}$; here $\chi_l$ is the Majorana fermion operator, index $l \in (1, N)$ and $N$ is the full number of sites available for fermions. Our aim is to calculate the energy dissipation rate $W(\omega)$ caused by the periodic modulation described by $\Phi(t)$. Below we will show that pumping at a certain frequency $\omega_P$ (its value will be specified below) can change the properties of the model crucially, so it is useful to consider $\Phi(t) = A\cos(\omega_P t) + f(t)$ and to study the energy dissipation rate due to the application of the field $f(t) \propto e^{-i\omega t}$. To begin with, we make several technical notes about our calculation and introduce few useful notations.

We use dimensionless time $x \equiv \frac{2\pi t}{\beta}$ and dimensional frequency $\Omega = \frac{\omega\beta}{2\pi}$. As we plan to study non-equilibrium properties of the model with strong interaction, we need to use Keldysh formalism [10]. The Green function of fermions is defined as $G_{s_1,s_0}(x_1, x_0) \equiv -i \sum_l \langle \chi_{l,s_1}(x_1) \chi_{l,s_0}(x_0) \rangle$, where $s_\alpha \in (+1, -1)$ denote upper or lower part of the Keldysh contour. The saddle-point Schwinger - Dyson equations read, for this model in the limit $N \gg 1$, $|x_1 - x_0| \gg \frac{T}{J}$ and at $\Phi = 0$:

$$\Sigma_{s_1,s_0}(x_1, x_0) = s_1 s_0 \left[ J^2 G_{s_1,s_0}^3(x_1, x_0) - \Gamma^2 G_{s_1,s_0}(x_1, x_0) \right], \qquad \Sigma \circ G = -\hat{1}. \tag{2}$$

Let us discuss several well-known properties of the original $SYK$ model ($\Gamma = 0$). For every solution $\hat{G}$ of Eqs.(2), the function

$$G_{s_1,s_0}^\phi(x_1, x_0) = G_{s_1,s_0}(\phi^{s_1}(x_1), \phi^{s_0}(x_0)) \left[ \phi^{s_1}{}'(x_1) \phi^{s_0}{}'(x_0) \right]^\Delta \tag{3}$$

is also solution; here $\phi_s$ is an arbitrary monotonous function and $\Delta = \frac{1}{4}$; we use this generalized notation as it will be useful below for regularization purpose. The translation invariant solution has the form: $\hat{G}(x_1, x_0) = -i \left( \frac{b}{J^2} \right)^\Delta \hat{g}(x_1 - x_0)$ where (see Ref. [11])

$$\hat{g}(x > 0) = \left( \frac{1}{4\sinh^2\left(\frac{x}{2}\right)} \right)^\Delta \begin{pmatrix} e^{-i\pi\Delta} & -e^{i\pi\Delta} \\ e^{-i\pi\Delta} & -e^{i\pi\Delta} \end{pmatrix} \tag{4}$$

and $b = 4\pi$. To find Green's function for $x_1 < x_0$ one can use the relation $\hat{G}(x_1, x_0)^T = -\hat{G}(x_0, x_1)$. The symmetry group $SL(2, \mathbb{R})$ of this solution is smaller than the full symmetry group respected by the equations (2); it results in the appearance of a soft reparametrization mode and strong influence of its fluctuations. To take them into account, we replace each Green function entering the action by $G^\phi$ and then integrate over $\phi(x)$ with the following action [3–5,12]:

$$S = S_{SYK} + S_2, \qquad S_{SYK} = -\varepsilon_0 \sum_s \int \{e^{\phi^s(x)}, x\} dx,$$

$$S_2 = \frac{ig}{2\varepsilon_0} \sum_{s_1,s_0} \int dx_1 dx_0 s_1 s_0 \left[ g_{s_1,s_0}^\phi(x_1, x_0) \right]^2 (1 + \Phi^{s_1}(x_1))(1 + \Phi^{s_0}(x_0)),$$

$$g_{s_1,s_0}^\phi(x_1, x_0) = g_{s_1,s_0}(\phi^{s_1}(x_1) - \phi^{s_0}(x_0)) \left[ \phi^{s_1}{}'(x_1) \phi^{s_0}{}'(x_0) \right]^\Delta, \tag{5}$$

where $\{Y, X\}$ denotes Schwarzian derivate and parameters of the action are given by

$$\varepsilon_0 = \frac{2\pi\gamma}{\beta J}, \quad g = \frac{N\sqrt{b}\gamma\Gamma^2}{2J^2}, \quad \gamma = \alpha_S N, \quad \alpha_s \approx 0.05. \tag{6}$$

The action (5) describes general problem with non-zero $\Gamma$ and parametric modulation $\Phi(x)$, The magnitudes of $\Phi^{(+,-)}(x) = A\cos(\Omega_P x) + f^{+,-}(x)$ at the upper and the lower branches of the Keldysh contour can be different, we denote their difference as "quantum component" $f^+(x) - f^-(x) = f^q(x)$ which is the source field useful for the calculation of susceptibility; classical component is defined as $f^{cl} = \frac{1}{2}(f^+(x) + f^-(x))$.

The rate of energy dissipation $W(\omega)a_\omega^2$ due to the presence of oscillating field $f^{cl}(t) = a_\omega\cos(\omega t)$ can be calculated with the help of this action in the following way:

$$W(\Omega) = \frac{\Omega}{2}\Im\chi(\Omega), \qquad \chi(\Omega) = -\frac{i}{2}\frac{\delta^2 Z_\Phi}{\delta f_\Omega^q \delta f_\Omega^{cl}}, \qquad Z_\Phi = \int \mathcal{D}\phi\, e^{iS}. \tag{7}$$

In the previous work [1] we have shown that at $T \ll \Gamma$ fluctuations of the soft mode are suppressed, while modification of the saddle point solution is small as long as $T \gg \Gamma^2/J$. In a broad intermediate range $\Gamma^2/J \ll T \ll \Gamma$ considered in the following calculation, we can assume that $\phi^s(x) = x + u^s(x)$ where $u^{s\prime}(x) \ll 1$. Thus we can work in the Gaussian approximation over $u(x)$ and the action (5) take the following form:

$$S = S_{SYK} + S_{\tilde\Phi}, \qquad S_{\tilde\Phi} = S_{\tilde\Phi}^{(0)} + S_{\tilde\Phi}^{(1)} + S_{\tilde\Phi}^{(2)},$$

$$S_{SYK} = \frac{1}{2}\int \frac{d\Omega}{2\pi}\left[\hat{\mathcal{G}}_0(\Omega)\right]^{-1}_{s_1, s_2} u^{s_1}_{-\Omega} u^{s_2}_{\Omega}, \qquad S_{\tilde\Phi}^{(0)} = i\frac{g}{2\varepsilon_0}\int\frac{d\Omega}{2\pi}L^{(0)}_{s_1, s_2}(\Omega)\tilde\Phi^{s_1}_{-\Omega}\tilde\Phi^{s_2}_{\Omega},$$

$$S_{\tilde\Phi}^{(1)} = i\frac{g}{2\varepsilon_0}\int\frac{d\Omega_0 d\Omega_1}{(2\pi)^2}L^{(1)}_{s_1, s_2, s_3}(\Omega_0, \Omega_1)u^{s_1}_{\Omega_0}\tilde\Phi^{s_2}_{-\Omega_1 - \frac{\Omega_0}{2}}\tilde\Phi^{s_3}_{\Omega_1 - \frac{\Omega_0}{2}},$$

$$S_{\tilde\Phi}^{(2)} = i\frac{g}{2\varepsilon_0}\int\frac{d\Omega_0 d\Omega_1 d\Omega_2}{(2\pi)^3}L^{(2)}_{s_1, s_2, s_3, s_4}(\Omega_0, \Omega_1, \Omega_2)u^{s_1}_{\Omega_0}u^{s_2}_{\Omega_1}\tilde\Phi^{s_3}_{-\Omega_2 - \frac{\Omega_0 + \Omega_1}{2}}\tilde\Phi^{s_4}_{\Omega_2 - \frac{\Omega_0 + \Omega_1}{2}}. \tag{8}$$

Here we have introduced new notations $\tilde\Phi^\pm(x) = 1 + \Phi^\pm(x)$ and $\hat{L}^{(i)}$ are tensors obtained from the original action (see Appendices A,B). This action is Gaussian in terms of fluctuations of $u(x)$ so we can easily do the functional integral in Eq.(7). The propagator of $u(x)$ in this model is determined by the quadratic over $u(x)$ form in following action:

$$S_0 = S_{SYK} + S_{\tilde\Phi_0}^{(2)}, \tag{9}$$

where $S_{\tilde\Phi_0}^{(2)}$ denotes the term $S_{\tilde\Phi}^{(2)}$ from Eq.(8), evaluated at $\tilde\Phi_0^\pm = \tilde\Phi^\pm|_{f=0} = 1 + A\cos\Omega_P x$. It is useful to understand the functional $Z_\Phi$ defined in Eq.(7) as $Z_\Phi = \langle e^{i(S-S_0)}\rangle_0$ where $\langle\ldots\rangle_0$ means average with action $S_0$; the difference $S - S_0$ contains source terms needed to calculate susceptibility. Let us analyze the propagator corresponding to this action.

## 3 Fluctuation propagator without pumping.

The action of the soft mode was calculated in Apendices A,B,C and has the form

$$S_0 = \int\frac{d\Omega}{4\pi}\hat{u}^T_{-\Omega}\begin{pmatrix} 0 & \left[\mathcal{G}^A(\Omega)\right]^{-1} \\ \left[\mathcal{G}^R(\Omega)\right]^{-1} & -\mathcal{G}^K(\Omega)\left[\mathcal{G}^A(\Omega)\right]^{-1}\left[\mathcal{G}^R(\Omega)\right]^{-1} \end{pmatrix}\hat{u}_\Omega, \tag{10}$$

where

$$\left[\mathcal{G}^R(\Omega)\right]^{-1} = \Omega^2\left(\varepsilon_0\left(\Omega^2 + 1\right) - \frac{g}{2\varepsilon_0}\psi(-\Omega)\right),$$

$$-\mathcal{G}^K(\Omega)\left[\mathcal{G}^A(\Omega)\right]^{-1}\left[\mathcal{G}^R(\Omega)\right]^{-1} = i\frac{g}{2\varepsilon_0}\pi\Omega^2, \tag{11}$$

$$\psi(\Omega) = \Psi\left(\tfrac{1}{2} + i\Omega\right) - \Psi\left(-\tfrac{1}{2}\right), \quad \Psi(z) = \partial_z\ln\Gamma(z).$$

The action (10) is non-local, thus $[\mathcal{G}^R]^{-1}$ has non-zero imaginary part; as a result, the distribution function $F(\Omega)$ can be determined by the standard relation $\mathcal{G}^K(\Omega) \equiv F(\Omega)\big(\mathcal{G}^R(\Omega) - \mathcal{G}^A(\Omega)\big)$. We find then $F(\Omega) = \coth(\pi\Omega)$, corresponding to the equilirium bosonic distribution in the absence of pumping.

For $g \gg \varepsilon_0^2 \Leftrightarrow T \ll \Gamma$ the bosonic Green function $\mathcal{G}(\Omega)$ corresponding to the action (10,12) demonstrates resonant behavior with the frequencies $\pm\Omega_R$ and the resonance width $\Omega_W$. These parameters are given by:

$$\Omega_R^2 = \frac{g}{2\varepsilon_0^2} \ln(\Omega_R)\,, \qquad \frac{\Omega_W}{\Omega_R} \approx \frac{\pi}{4\ln(\Omega_R)}\,, \qquad \frac{g}{2\varepsilon_0^2} = \frac{\sqrt{b}}{\alpha_S(4\pi)^2}\left(\frac{\Gamma}{T}\right)^2\,. \tag{12}$$

The frequency $\Omega_R$ corresponds to the lowest level-spacing in the polaron problem studied in Ref. [1], in the case of many polaron levels (corresponding to large $\kappa$ parameter defined in Ref. [1]). It is interesting to note that parameters of this resonance peak are detemined by the ratio $\frac{\Gamma}{T}$ only, while the largest energy scale of the problem $J$ does not enter here. Below we will be interested in the low-$T$ region, $T \ll \Gamma$, where deviations from pure SYK model are most substantial. Note that fermionic Green function $G(\epsilon)$ is only weakly (by factors $\propto 1/N$) modified in the same parameter range, being close to the "conformal limit" solution, while bosonic Green function $\mathcal{G}(\Omega)$ strongly depends on $\Gamma$ and $T$.

Well-defined resonance behaviour with "quality factor" $\ln(\Omega_R) \sim \ln\frac{\Gamma}{T} \gg 1$ is rather surprising to find in our problem which lacks any apparent energy scale determining the frequency of this resonance. While $\Gamma$ itself is just the width of the "single-particle band" that occurs due to $\Gamma_{ij}$ matrix elements, the frequency $\omega_R$ and width $\omega_W$ of the resonance are given, in physical units by

$$\omega_R = 2\pi T\Omega_R = \frac{\Gamma}{4\alpha_s\sqrt{\pi}} \ln^{1/2}\frac{\Gamma}{T}\,, \tag{13}$$

$$\omega_W = 2\pi T\Omega_W = \frac{\sqrt{\pi}}{16\alpha_s}\frac{\Gamma}{\ln^{1/2}\frac{\Gamma}{T}}\,. \tag{14}$$

They are larger (correspondingly, smaller) than $\Gamma$ by the factor $\sqrt{\ln\frac{\Gamma}{T}}$, coming from strong interaction between fermions. Note that this result is specific for $\mathrm{SYK}_4$ model of interaction. For any $\mathrm{SYK}_{2k}$ model with $k > 2$ similar derivation would lead to some expressions of the type of Eqs.(10,12) with the function $\psi(\Omega)$ replaced by some power-law function $\tilde{\psi}(\Omega) \sim \Omega^\alpha$ where $\alpha = 4/k - 1$; as a result the ratio $\frac{\Omega_W}{\Omega_R} \propto \frac{\Im\psi(\Omega_0)}{\Re\psi(\Omega_0)} \sim O(\alpha)$ would be of the order of unity. The $\mathrm{SYK}_4$ model is special since it leads, effectively, to $\alpha = 0$. The resonant behaviour leads to a substantial change of the dissipation rate compared to the original $\mathrm{SYK}_4$ model. In addition, the system properties can be seriously modified by applying $ac$ pumping with frequency $\omega_P \approx \omega_R$.

## 4 Dissipation rate in the linear regime.

To find susceptibility $\chi(\Omega)$ we need to calculate the average $[S^{(1)}]^2$, while the term $S^{(2)}$ can be neglected. The result reads:

$$\chi(\Omega) = \frac{1}{2}\frac{\delta^2}{\delta f_\Omega^q \delta f_\Omega^{cl}}\left(S_{\tilde{\Phi}}^{(0)} + \frac{i}{2}\langle[S^{(1)}]^2\rangle\right)\,. \tag{15}$$

Let us first discuss the case of pure SYK model (that is, $\Gamma = 0$); then susceptibility is determined by the first term of (15) only:

$$\chi_{SYK}(\Omega) = \frac{2g}{\varepsilon_0}\left(\ln\left(\frac{\beta J}{2\pi}\right) - \psi(-\Omega)\right)\,. \tag{16}$$

To get dissipative part of physical susceptibility, we need also to multiply $\Im\chi_{SYK}(\Omega)$ by the factor $2\pi T = \omega/\Omega$; the result is (we used also relations (6)):

$$\Im\chi_{SYK}(\omega) = 2\pi T \frac{\pi g}{\varepsilon_0}\tanh\frac{\omega}{2T} = \pi^{3/2}\frac{N\Gamma^2}{J}\tanh\frac{\omega}{2T}. \tag{17}$$

The logarithmically large term in Eq.(16) comes due a formal divergence in the integral over $dx_1$ in the term $S_2$ in the action (5). This divergence is present since $g(x_1, x_0) \propto |x_1 - x_0|^{-1}$ at $x_1 \to x_0$. We need to cut-off this integral at the scale $[\Delta x]_{min} \sim \frac{J}{T}$ since at lower $\Delta x$ our long-wavelength expression for the action is not valid. Below we will omit this term from the expressions for susceptibility, since it is just real constant independent on $\Omega$. The result (17) is known in the theory of non-Fermi liquid [13, 14]; it is also related to linear dependence of resistance on temperature in the model of Ref. [11].

Inclusion of quadratic terms $\sim \Gamma_{ij}$ into the Hamiltonian leads to strong modification of the $u(x)$ fluctuation propagator, and thus to a considerable change in susceptibility. Detailed calculation of $\frac{i}{2}\langle[S^{(1)}]^2$ can be found in the appendix D. After some algebra, one can find susceptibility in the following form:

$$\chi(\Omega) = -\frac{2g}{\varepsilon_0}\psi(-\Omega)\left[1 + \frac{g}{2\varepsilon_0}\Omega^2\mathcal{G}_\Omega^R\psi(-\Omega)\right]. \tag{18}$$

This formula can be understood in the following way. Let us consider a representation for the $S_2$ from Eq.(5) using Keldysh contour $\mathcal{C}$:

$$S_2 = i\frac{g}{2\varepsilon_0}\int_{\mathcal{C}} dx_1 dx_0\left[g^\phi(x_1, x_0)\right]^2(1 + \Phi(x_1))(1 + \Phi(x_0)). \tag{19}$$

Now we consider arbitrary re-parametrization of this contour $x \mapsto \tilde{x}(x)$. We can rewrite this part of the action using the re-parametrization as follows:

$$S_2 = i\frac{g}{2\varepsilon_0}\int_{\mathcal{C}} d\tilde{x}_1 d\tilde{x}_0\left[g^\phi(\tilde{x}_1, \tilde{x}_0)\right]^2(1 + \Phi(\tilde{x}_1))(1 + \Phi(\tilde{x}_0))\left[\frac{\partial\tilde{x}_1}{\partial x_1}\frac{\partial\tilde{x}_0}{\partial x_0}\right]^{2\Delta-1}. \tag{20}$$

Now we can fix the choice of re-parametrization as the solution of the following equation: $(1 + \Phi(\tilde{x}))\left[\frac{\partial\tilde{x}}{\partial x}\right]^{2\Delta-1} = 1$. It will simplify the action and leads to the in the form: $S_2 = i\frac{g}{2\varepsilon_0}\int_{\mathcal{C}} d\tilde{x}_1 d\tilde{x}_0\left[g^\phi(\tilde{x}_1, \tilde{x}_0)\right]^2$. As a result we will find the action written in the term of the function $\tilde{\phi}(\tilde{x})$. In the linear approximation we can write:

$$\phi(\tilde{x}) \approx \tilde{x} + u(x) \approx x + u(x) + \frac{1}{1-2\Delta}\int_{-\infty}^{x}\Phi(x')dx'. \tag{21}$$

It means that we can write the quadratic action and, as a result, the effective action for the sources in the form:

$$S_{SYK} = \frac{1}{2}\int\frac{d\Omega}{2\pi}\left\{u_\Omega^\dagger\left[\hat{\mathcal{G}}_0(\Omega)\right]^{-1}u_\Omega - \left(u_\Omega + \frac{2i}{\Omega}\Phi_\Omega\right)^\dagger\hat{\Sigma}(\Omega)\left(u_\Omega + \frac{2i}{\Omega}\Phi_\Omega\right)\right\},$$

$$S_{eff} = \frac{1}{2}\int\frac{d\Omega}{2\pi}\frac{4}{\Omega^2}\Phi_\Omega^\dagger\left\{\hat{\Sigma}(\Omega) + \hat{\Sigma}(\Omega)\hat{G}(\Omega)\hat{\Sigma}(\Omega)\right\}\Phi_\Omega,$$

$$\hat{\Sigma}(\Omega) = \frac{g}{2\varepsilon_0}\Omega^2\begin{pmatrix}0 & \psi(\Omega) \\ \psi(-\Omega) & -i\pi\end{pmatrix}. \tag{22}$$

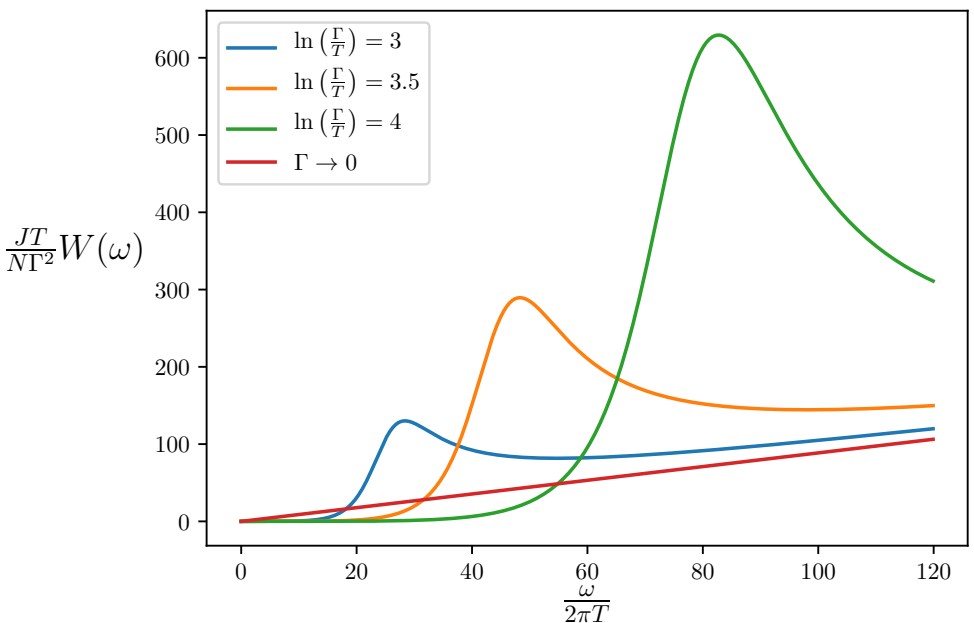

Figure 1: Dissipation rate at large $\Gamma/T$ and at $\Gamma \to 0$; resonant enhancement is seen at $\omega \approx \omega_R$. Note that low-frequency part of the figure with strong suppression of $W(\omega)$ corespond to the contribution of the soft mode only; full magnitude of the rate should include here also the contribution from Eq.(26).

The special structure $\Sigma + \Sigma G \Sigma$ in the action $S_{eff}$ leads to the relation (18) between observable susceptibility $\chi(\Omega)$ and bosonic Green function $G(\Omega)$. We note that the above consideration is not applicable to a ultraviolet-singular part of the susceptibility $\ln(\beta J/2\pi)$, but it is not important for our analysis since no contribution to $\Im\chi(\Omega)$ comes from the ultraviolet.

The effect of quadratic perturbations is best represented by the imaginary part of susceptibility:

$$\Im\chi(\Omega) = \frac{\left(\Omega^2 + 1\right)^2 \Im\chi_{SYK}(\Omega)}{\left(\Omega^2 + 1 - \frac{g}{2\varepsilon_0^2}\Re\psi(-\Omega)\right)^2 + \left[\frac{g}{2\varepsilon_0^2}\Im\psi(-\Omega)\right]^2} . \tag{23}$$

Under the condition $\Gamma \gg T \Leftrightarrow g \gg \varepsilon_0^2$, formula (23) demonstrates resonance peak at $\Omega \approx \Omega_R$. At high frequencies $\Omega \gg \Omega_R$ new result (23) reduces to pure SYK one. Near resonance, at $\Omega \approx \Omega_R$, dissipation in our model is enhanced by the factor

$$\frac{\Im\chi(\Omega)}{\Im\chi_{SYK}(\Omega)} = \left(\frac{\Omega_R}{2\Omega_W}\right)^2 = \frac{4}{\pi^2}\ln^2\Omega_R \gg 1. \tag{24}$$

Frequency-dependence of dissipation rate $W(\omega)$ corresponding to Eq.(23) is shown in Fig.(1) for several large values of $\Gamma/T$ ratio.

At low frequencies $\Omega \ll \Omega_W$ dissipation given by Eq.(23) is strongly suppressed w.r.t. pure SYK model:

$$\frac{\Im\chi(\Omega)}{\Im\chi_{SYK}(\Omega)} = \left(\frac{\Omega^2 + 1}{2\Omega_R\Omega_W|\psi(-\Omega)|}\right)^2 \sim \left(\frac{(2\pi T)^2 + \omega^2}{\Gamma^2}\right)^2 . \tag{25}$$

Applicability of the result (25) is limited since it is obtained for the temperature region $T \ll \Gamma$ without account for modification of the saddle-point solution which is known to lead to Fermi-liquid saddle point at $T \leq T_{FL} = \Gamma^2/J$. In other terms, in the above calculation we did not

account for the hard modes which control the crossover to the FL state. Such a calculation in provided in the appendix E, and it results in the following additional term to dissipative part of susceptibility written below for $\omega \ll T$:

$$\Im \chi_{\text{add}}(\omega) = \mathcal{C} \, \omega N \mathcal{M}^2 = \frac{\mathcal{C}}{(4\pi)^2} \, \omega N \left( \frac{T_{FL}}{T} \right)^2, \tag{26}$$

where numerical coefficient $\mathcal{C} \approx 234$. At $T \sim T_{FL}$ the contribution (26) is of the same order of magnitude as the Fermi-liquid result $\Im \chi_{FL}(\omega) \sim N\omega$. The contribution (26) becomes comparable to the soft-mode contribution (25) at $T \approx T_{cr} = \Gamma(\Gamma/J)^{1/5}$. Thus we see that Eq.(25) is valid in the narrow range $\Gamma(\Gamma/J)^{1/5} < T < \Gamma$.

## 5 Nonlinear pumping effects.

We consider now the behavior of our system at non-equilibrium conditions, under the external *ac* pumping with frequency $\omega_P \approx \omega_R$ and amplitude *A*, thus $\Phi(t) = A\cos(\omega_P t) + f(t)$. The difference $|\omega_P - \omega_R|$ is irrelevant as long as it is much smaller than resonance width $\omega_W$, and below we will assume $\omega_P = \omega_R$. We are interested here in the modification of low-frequency response at $\omega \ll \omega_W$ due to such a pumping.

Important remark is in order: pumping leads to energy absorption by our system, thus for the stationary distribution to exist, some kind of coupling to external bath is necessary. Here we prefer to employ another approach: we consider finite-time pumping during timescale $t_{\text{pump}}$ chosen in such a way that the total energy absorbed in our system $E_{\text{abs}} = W(\omega)A^2 t_{\text{pump}}$ is small enough, so the increase $\Delta T$ of its temperature is relatively small, $\Delta T \ll T$. In this respect our approach is very different from the one developed in Ref. [15] for the $N = \infty$ limit of the combined SYK$_4$+SYK$_2$ system: they studied the limit of strong energy pumping. Then we can study quasi-stationary response of our system at frequencies $\omega$ restricted by the condition $\omega \gg 1/t_{\text{pump}}$. Increase of the temperature due to external pumping source is $\Delta T = E_{\text{abs}}/C(T)$ where $C(T)$ is the heat capacity. The major contribution to $C(T)$ in the interesting range $T \ll \Gamma$ comes from quadratic terms in the action. Heat capacity $C(T)$ can be calculated in the saddle-point approximation, see appendices B.3 and E; the corresponding calculation of the entropy was performed in [11], but it was left unnoticed that dominant at $T < \Gamma$ contribution to $C(T)$ grows as $\sqrt{4\pi}N\Gamma^2/JT$ before reaching the maximum at $T \approx \Gamma^2/J$ $C(T) \approx \sqrt{4\pi}N\Gamma^2/JT$. In result, we find sequence of inequlities for magnitudes of *A* and $\omega$:

$$\omega_R A^2 \ln^2 \frac{\omega_R}{T} \ll \frac{1}{t_{\text{pump}}} \ll \omega. \tag{27}$$

Nonzero pumping amplitude affects low-frequency susceptibility in two ways: first, it changes the action of the soft mode; second, it creates a correction to the term $\frac{i}{2}\langle [S^{(1)}]^2 \rangle$ since $S^{(1)}$ is of the second order in $\tilde{\Phi}$. The second contribution occurs to be small at $\omega \ll \omega_W$, as it is shown in the appendix D.4, so we neglect it.

Pumping-induced corrections to the action of the soft mode comes in two ways. First contribution which we call "direct" one, is due to the terms $\sim A^2$ and contains terms like $u_{-\Omega}u_{\Omega}$ in the action. Second contribution is "indirect" in the sense that it is $\propto A$ and it contains mixture of high- and low-frequency harmonics, $u_{-\Omega}u_{\Omega+\Omega_R}$. After Gaussian integration over fast modes $u_{\Omega_f}$ with $\Omega_f \approx \Omega_R$, these terms also produce contribution to the action of slow soft modes. The combination of direct and indirect terms leads (for details see appendix C) to the additional action of the soft mode:

$$\delta_A S_{soft} = \pi \Omega_P \frac{ig}{2\varepsilon_0} \left( \frac{A}{2} \right)^2 \int \frac{d\Omega}{(2\pi)} \hat{u}_{-\Omega}^T \begin{pmatrix} 0 & -\Omega \\ \Omega & 4\Omega_P \end{pmatrix} \hat{u}_\Omega. \tag{28}$$

Off-diagonal terms in the $2 \times 2$ matrix in Eq.(28) are proportional to $\Omega$, contrary to the analogous terms in the original action $S_{soft}$, see Eq.(10), which starts from terms $\propto \Omega^2$. These linear in $\Omega$ terms indicates the appearence of *friction* due to non-equilibrium nature of the system under pumping. Apart from modification of quadratic in $u(x)$ terms, pumping leads also to the appearence of the singular contribution $\propto A^2 t_{pump}$ to the average value $\langle u(x) \rangle$ which reflects the raise of temperature $T \rightarrow T + \Delta T$ due to pumping, see appendix E. In the harmonic approximation over $u(x)$ we used, the average $\langle u(x) \rangle$ does not change fluctuation propagator, however it will be affected once non-linear in $u(x)$ terms will be taken into account. The condition (27) ensures that $\Delta T \ll T$ and this effect is small.

Combining two contributions to the action, Eqs.(10) and (28), we obtain full action for low-frequency soft mode fluctuations. The corresponding retarded Green function is

$$\left[ \mathcal{G}^R(\Omega) \right]^{-1} = \Omega^2 \left( \varepsilon_0 \left( \Omega^2 + 1 \right) - \frac{g}{2\varepsilon_0} \psi(-\Omega) \right) + \frac{i\pi g \Omega_P}{\varepsilon_0} \Omega \left( \frac{A}{2} \right)^2 . \tag{29}$$

Now we can use Eq.(18) to find susceptibility $\chi(\omega)$ with respect to *ac* probe field $f_\Omega$ in the range $\frac{1}{T t_{pump}} \ll \Omega \ll \Omega_R$, see also Eq.(27):

$$\Im \chi(\Omega) = \frac{\pi g}{\varepsilon_0} \left( \frac{\Omega_R}{\Omega} A^2 + \frac{(\Omega^2 + 1)^2}{|\frac{g}{2\varepsilon_0^2} \psi(-\Omega)|^2} \tanh(\pi \Omega) \right) . \tag{30}$$

Equation (30) demonstrates the presence of the typical scale $I(\Omega) = \frac{\Omega(\Omega^2+1)}{\Omega_R^2 \Omega_W}$ for the pumping amplitude $A$; the response frequency $\Omega \ll \Omega_W$. At $A \ll I(\Omega)$ pumping is weak and does not modify susceptibility and dissipation rate. In the range $I(\Omega) \ll A$ one finds $\Im \chi_1^A(\Omega) \approx \pi A^2 \frac{g}{\varepsilon_0} \frac{\Omega_R}{\Omega}$ and dissipation rate

$$W(\omega) = \frac{\omega}{2} \Im \chi(\omega) = \frac{A^2}{2} N \pi^{3/2} \omega_R \frac{\Gamma^2}{J} , \tag{31}$$

which is somewhat unusual: $W(\omega)$ does not depend on $\omega$ and $T$, which reminds a "dry friction" phenomenon. Fig. (2) represents $W(\omega)$ behavior as it follows from Eq.(30), for the magnitudes of pumping $A = 5 \cdot 10^{-3}$ and several small ratios $T/\Gamma \ll 1$.

# 6 Conclusion

We have studied energy absorption in a driven system of strongly correlated Mayorana fermions of mixed $SYK_4 + SYK_2$ type, with relatively weak quadratic part of the Hamiltonian, $\Gamma \ll J$, at the temperatures $T \gg T_{FL} = \Gamma^2/J$. Our major results refer to the region $T \leq \Gamma$ where new type of unviersal behaviour was found: fluctuations of the soft mode are characterized by the resonant behaviour with frequency $\omega_R$ and width $\omega_W$ given by Eqs.(13,14). Quality factor of this resonance is $Q = \frac{4}{\pi} \ln \frac{\Gamma}{T}$. Suprisingly, both $\omega_R$ and $\omega_W$ do not depend on the largest interaction energy scale $J \gg \Gamma$. The frequency $\omega_R$ directly corresponds to the polaron bound-state energy found in Ref. [1] via Matsubara-time approach. Although Fermionic Green function itself is accurately described by the conformal saddle-point solution, physical properties appears to be sensitive to the bosonic collective mode that becomes instrumental at $T < \Gamma$.

Physical significance of the polaron soft mode is demonstrated via the results we obtained for the energy dissipation due to parametric modulation of the quadratic part of the Hamiltonian. For near-resonance modulation frequencies $\omega \approx \omega_R$ it is found to be enhanced by the factor $\frac{1}{4} Q^2$ w.r.t. pure $SYK_4$ model, see Eqs.(23,24). On the contrary, at lower frequencies $\omega \ll \omega_R$ dissipation rate is suppressed, see Eqs.(25,26).

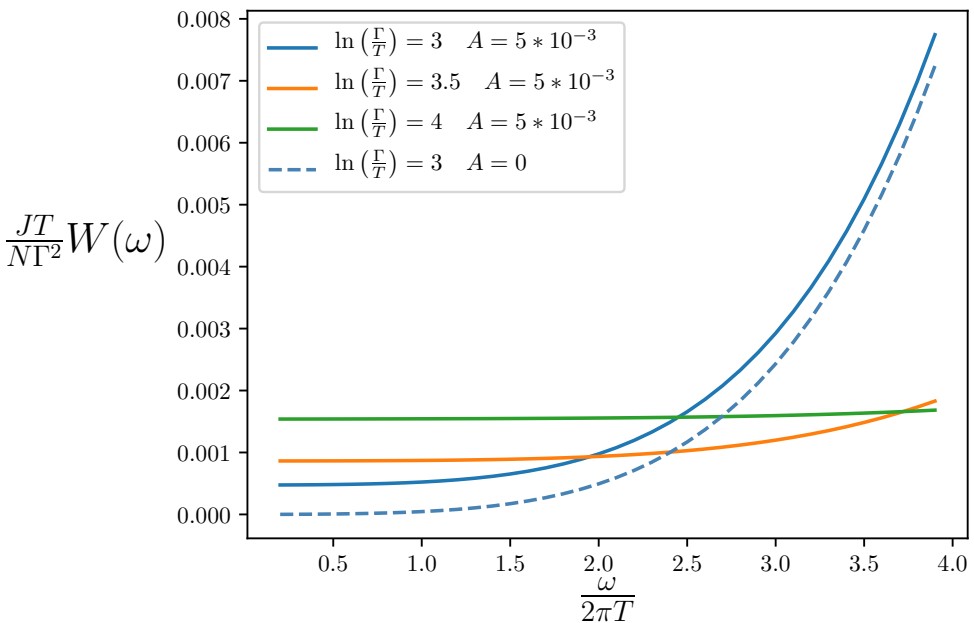

Figure 2: Dissipation rate at small frequencies $\omega \ll \omega_R$ in the presence of pumping with amplitude $A = 5 \cdot 10^{-3}$. Very weak $\omega$-dependence at intermediate frequencies correspond to the "dry friction domain", Eq.(31).

Pumping the system with *ac* modulation of finite amplitude $A$ at the resonant frequency $\omega_R$ leads to a non-equilibrium state those response to a linear low-$\omega$ perturbation differs considerably from the case of $A = 0$. First, the pumping leads to a "friction term" in the action of the soft mode, see Eqs.(28),(29). Secondly, the power $W(\omega)$ dissipated due to *ac* modulation is nearly $\omega$-independent in the range $I(\omega) \ll A^2$, see Fig.2.

In spatially extended systems the polaron soft mode is expected to be related to the energy transport. Rich behaviour of its propagator $\mathcal{G}(\omega)$ as function of $\Gamma$, $T$ and frequency indicates that transport properties of extended SYK-based models may occur to be more diverse than it seems to follow from the saddle-point analysis [11].

# Acknowledgements

We are grateful to A. Yu. Kitaev for numerous useful discussions and to K. S. Tikhonov for important comments.

**Funding information** Research of A.V.L. was partially supported by the Basis Foundation, by the Basic research program of the HSE and by the RFBR grant # 20-32-90057.

# A   The effective action

The action of the model is present the main text and has the following form:

$$S = S_{SYK} + S_2, \quad S_{SYK} = -\varepsilon_0 \sum_s \int \{e^{\phi^s(x)}, x\} dx,$$

$$S_2 = \frac{ig}{2\varepsilon_0} \sum_{s_1,s_0} \int dx_1 dx_0 s_1 s_0 \left[g_{s_1,s_0}^{\phi}(x_1,x_0)\right]^2 (1 + \Phi^{s_1}(x_1))(1 + \Phi^{s_0}(x_0)),$$

$$g_{s_1,s_0}^{\phi}(x_1,x_0) = g_{s_1,s_0}(\phi^{s_1}(x_1) - \phi^{s_0}(x_0)) \left[\phi^{s_1\prime}(x_1)\phi^{s_0\prime}(x_0)\right]^{\Delta},$$

$$\varepsilon_0 = \frac{2\pi\gamma}{\beta J}, \quad g = \frac{N\sqrt{b}\gamma\Gamma^2}{2J^2}, \quad \gamma = \alpha_S N, \quad \alpha_s \approx 0.05,$$

$$\hat{g}(x) = \left(\frac{1}{4\sinh^2\left(\frac{x}{2}\right)}\right)^{\Delta} \left[\theta(x)\begin{pmatrix} e^{-i\pi\Delta} & -e^{i\pi\Delta} \\ e^{-i\pi\Delta} & -e^{i\pi\Delta} \end{pmatrix} + \theta(-x)\begin{pmatrix} -e^{-i\pi\Delta} & -e^{-i\pi\Delta} \\ e^{i\pi\Delta} & e^{i\pi\Delta} \end{pmatrix}\right]. \tag{32}$$

Our aim is to calculate susceptibility with respect to the field $\Phi$, which is defined as follows:

$$\chi^R(t - t') = -\frac{i}{2}\frac{\delta^2 Z_\Phi}{\delta\Phi^q(t)\delta\Phi^{cl}(t')}, \qquad Z_\Phi = \int \mathcal{D}\phi \, e^{iS}. \tag{33}$$

We consider the limit of weak fluctuations, thus the action can be written in the form

$$S = S_{SYK} + S_{\tilde{\Phi}}, \quad S_{\tilde{\Phi}} = S_{\tilde{\Phi}}^{(0)} + S_{\tilde{\Phi}}^{(1)} + S_{\tilde{\Phi}}^{(2)}, \quad S_{SYK} = \frac{1}{2}\int \frac{d\Omega}{2\pi}\left[\hat{\mathcal{G}}_0(\Omega)\right]_{s_1,s_2}^{-1} u_{-\Omega}^{s_1} u_{\Omega}^{s_2},$$

$$S_{\tilde{\Phi}}^{(0)} = i\frac{g}{2\varepsilon_0}\int \frac{d\Omega}{2\pi} L_{s_1,s_2}^{(0)}(\Omega)\tilde{\Phi}_{-\Omega}^{s_1}\tilde{\Phi}_{\Omega}^{s_2}, \quad S_{\tilde{\Phi}}^{(1)} = i\frac{g}{2\varepsilon_0}\int \frac{d\Omega_0 d\Omega_1}{(2\pi)^2} L_{s_1,s_2,s_3}^{(1)}(\Omega_0,\Omega_1)u_{\Omega_0}^{s_1}\tilde{\Phi}_{-\Omega_1-\frac{\Omega_0}{2}}^{s_2}\tilde{\Phi}_{\Omega_1-\frac{\Omega_0}{2}}^{s_3},$$

$$S_{\tilde{\Phi}}^{(2)} = i\frac{g}{2\varepsilon_0}\int \frac{d\Omega_0 d\Omega_1 d\Omega_2}{(2\pi)^3} L_{s_1,s_2,s_3,s_4}^{(2)}(\Omega_0,\Omega_1,\Omega_2)u_{\Omega_0}^{s_1}u_{\Omega_1}^{s_2}\tilde{\Phi}_{-\Omega_2-\frac{\Omega_0+\Omega_1}{2}}^{s_3}\tilde{\Phi}_{\Omega_2-\frac{\Omega_0+\Omega_1}{2}}^{s_4}. \tag{34}$$

Here we have introduced $\tilde{\Phi}^{\pm}(x) = 1 + \Phi^{\pm}(x)$ and $\hat{L}^{(i)}$ are tensors which we will obtain below using Taylor expansion over $S_2$. Using these expressions we will analyze modified new propagator of soft modes and calculate the dissipation rate $W(\omega) = \frac{\omega}{2}\Im\chi^R(\omega)$.

# B   The Taylor expansion in powers of $S_2$.

## B.1   Frequency domain

We will start our analysis from representation of $s_1 s_0 \left[g_{s_1,s_0}(x)\right]^2$:

$$s_1 s_0 \left[g_{s_1,s_0}(x)\right]^2 = \left(\frac{1}{4\sinh^2\left(\frac{x}{2}\right)}\right)^d \left[\theta(x)\begin{pmatrix} e^{-i\pi d} & -e^{i\pi d} \\ -e^{-i\pi d} & e^{i\pi d} \end{pmatrix} + \theta(-x)\begin{pmatrix} e^{-i\pi d} & -e^{-i\pi d} \\ e^{i\pi d} & -e^{i\pi d} \end{pmatrix}\right]_{s_1,s_0}$$

$$= \int \frac{d\Omega}{2\pi}e^{-i\Omega x}\hat{L}_{s_1,s_0}(\Omega),$$

$$\hat{L}(\Omega) = \left[K_d(\Omega)\begin{pmatrix} e^{-i\pi d} & -e^{i\pi d} \\ -e^{-i\pi d} & e^{i\pi d} \end{pmatrix} + K_d(-\Omega)\begin{pmatrix} e^{-i\pi d} & -e^{-i\pi d} \\ -e^{-i\pi d} & e^{i\pi d} \end{pmatrix}\right],$$

$$K_d(\Omega) = \int_0^\infty \frac{dx}{2\pi}e^{-i\Omega x}\left(\frac{1}{4\sinh^2\left(\frac{x}{2}\right)}\right)^d = \frac{\Gamma(1-2d)\Gamma(d-i\Omega)}{\Gamma(1-d-i\Omega)}. \tag{35}$$

Here we have introduced $d = 2\Delta \to \frac{1}{2} - 0$ we need this parameter for proper limit as we see that $K_d(\Omega)$ is divergence at $d = \frac{1}{2}$. The matrix $L(\Omega)$ plays the crucial role in our calculation.

Using above expression we can write the terms from $S_2$ contained dependence on $u$ where $u$ is defined as $\phi^{\pm}(x) = x + u^{\pm}(x)$:

$$
\begin{aligned}
s_1 s_0 \left[ g^{\phi}_{s_1,s_0}(x_1, x_0) \right]^2 &= s_1 s_0 \left[ g_{s_1,s_0}(\phi^{s_1}(x_1) - \phi^{s_0}(x_0)) \right]^2 \left( \phi^{s_1\prime}(x_1) \phi^{s_0\prime}(x_0) \right)^d \\
&= \int \frac{d\Omega}{2\pi} e^{-i\Omega(\phi^{s_1}(x_1) - \phi^{s_0}(x_0))} \left( \phi^{s_1\prime}(x_1) \phi^{s_0\prime}(x_0) \right)^d \hat{L}_{s_1,s_0}(\Omega) \\
&= \int \frac{d\Omega}{2\pi} e^{-i\Omega(x_1 - x_0)} \hat{L}_{s_1,s_0}(\Omega) \left( (1 + u^{s_1\prime}(x_1))(1 + u^{s_0\prime}(x_0)) \right)^d \\
&\quad \times e^{-i\Omega(u^{s_1}(x_1) - u^{s_0}(x_0))}.
\end{aligned}
\tag{36}
$$

The last line in (36) is useful to develop the Taylor series over $u(x)$.

### B.2 Three orders of expansion over $u(x)$

For further calculations, we consider the following Taylor expansion:

$$
\begin{aligned}
\left( (1 + u^{s_1\prime}(x_1))(1 + u^{s_0\prime}(x_0)) \right)^d e^{-i\Omega(u^{s_1}(x_1) - u^{s_0}(x_0))} &= \\
F^{(0)}_{s_1,s_0}(x_1, x_2) + F^{(1)}_{s_1,s_0}(x_1, x_2) + F^{(2,1)}_{s_1,s_0}(x_1, x_2) + F^{(2,2)}_{s_1,s_0}(x_1, x_2), &\\
F^{(0)}_{s_1,s_0}(x_1, x_2) = 1, \qquad F^{(1)}_{s_1,s_0}(x_1, x_2) = d(u^{s_1\prime}(x_1) + u^{s_0\prime}(x_0)) - i\Omega(u^{s_1}(x_1) - u^{s_0}(x_0)), &\\
F^{(2,1)}_{s_1,s_0} = \frac{d(d-1)}{2} \left( [u^{s_1\prime}(x_1)]^2 + [u^{s_0\prime}(x_0)]^2 \right) - \frac{\Omega^2}{2} \left( [u^{s_1}(x_1)]^2 + [u^{s_0}(x_0)]^2 \right) & \\
- i\Omega d(u^{s_1}(x_1) u^{s_1\prime}(x_1) - u^{s_0}(x_0) u^{s_0\prime}(x_0)), &\\
F^{(2,2)}_{s_1,s_0} = d^2 u^{s_1\prime}(x_1) u^{s_0\prime}(x_0) + \Omega^2 u^{s_1}(x_1) u^{s_0}(x_0) - i\Omega d(u^{s_1}(x_1) u^{s_0\prime}(x_0) - u^{s_0}(x_0) u^{s_1\prime}(x_1)). &
\end{aligned}
\tag{37}
$$

We have divided the contribution proportional to $u^2$ into two different parts, $F^{(2,1)}$ and $F^{(2,2)}$. Each term in $F^{(2,1)}$ depends on single variable $u(x_i)$ whereas every term in $F^{(2,2)}$ depends on both $u(x_0)$ and $u(x_1)$. The expressions for $F^{(a)}$ enters the corresponding terms in the action, $S^{(i)}$:

$$
S^{(a)}_{\tilde{\Phi}} = \frac{ig}{2\varepsilon_0} \sum_{s_1,s_0} \int dx_1 dx_0 \int \frac{d\Omega}{2\pi} e^{-i\Omega(x_1 - x_0)} \hat{L}_{s_1,s_0}(\Omega) F^{(a)}_{s_1,s_0}(x_1, x_2) \tilde{\Phi}^{s_1}(x_1) \tilde{\Phi}^{s_0}(x_0),
\tag{38}
$$

where second-order term is composed of two parts:

$$
S^{(2)}_{\tilde{\Phi}} = S^{(2,1)}_{\tilde{\Phi}} + S^{(2,2)}_{\tilde{\Phi}}.
\tag{39}
$$

The terms $S^{(a)}_{\tilde{\Phi}}$ with $a = 1, 2$ will be calculated below; here we transform the Keldysh matrix form of term $S^{(0)}$ into more convenient representation via classical (cl) and quantum (q) components:

$$
S^{(0)}_{\tilde{\Phi}} = \frac{ig}{\varepsilon_0} \int \frac{d\Omega}{2\pi} \hat{\tilde{\Phi}}^T_{-\Omega} \hat{L}(\Omega) \hat{\tilde{\Phi}}_{\Omega}, \quad \hat{\tilde{\Phi}}_{\Omega} = \begin{pmatrix} \tilde{\Phi}^{cl}_{\Omega} \\ \tilde{\Phi}^{q}_{\Omega} \end{pmatrix}, \quad \tilde{\Phi}^{\pm} = \tilde{\Phi}^{cl} \pm \tilde{\Phi}^q,
\tag{40}
$$

$$
L(\Omega) = -2i \begin{pmatrix} 0 & K(-\Omega)\sin(\pi d) \\ K(\Omega)\sin(\pi d) & i\cos(\pi d)(K(-\Omega) + K(\Omega)) \end{pmatrix}.
$$

Eq.(40) provides us with convenient form of $S^{(0)}$ to be used below. In the following two subsections we calculate $S^{(1)}$ and $S^{(2)}$.

This expression also helps us to calculate contribution to the heat capacity from the perturbation on the mean-field level. The expression for the heat capacity takes the form:

$$
C = C_{SYK} + \frac{\partial}{\partial T} \left( \frac{1}{2} \frac{\delta S^{(0)}_{\tilde{\Phi}}}{\delta \Phi^q(t)} |_{\Phi=0} \right) = (2\pi)^2 \alpha_S N \frac{T}{J} + \frac{\partial}{\partial T} \frac{2\pi}{\beta} \frac{ig}{2\varepsilon_0} L_{q,cl}(0).
\tag{41}
$$

Taking the limit $d \to \frac{1}{2}$ we meet divergence which is present since $g^2(x) \propto |x|^{-1}$ at $x \to 0$. We need to cut-off this integral at the scale $x \propto \frac{T}{J}$. As a result we have:

$$C = (2\pi)^2 \alpha_S N \frac{T}{J} - \frac{\partial}{\partial_T} \frac{2\pi}{\beta} \frac{g}{\varepsilon_0} \ln(\beta J) = (2\pi)^2 \alpha_S N \frac{T}{J} + \sqrt{b} \frac{N\Gamma^2}{2TJ}. \tag{42}$$

### B.3   Calculation of $S^{(1)}$

To calculate $S^{(1)}$ we represent it in the form

$$
\begin{aligned}
S^{(1)}_{\check{\Phi}} =& \frac{ig}{2\varepsilon_0} \sum_{s_1,s_0} \int dx_1 dx_0 \int \frac{d\Omega_1}{2\pi} e^{-i\Omega_1(x_1-x_0)} \hat{L}_{s_1,s_0}(\Omega_1)\big(du^{s_1\prime}(x_1) - i\Omega_1 u^{s_1}(x_1)\big) \tilde{\Phi}^{s_1}(x_1)\tilde{\Phi}^{s_0}(x_0) \\
&+ \frac{ig}{2\varepsilon_0} \sum_{s_1,s_0} \int dx_1 dx_0 \int \frac{d\Omega_1}{2\pi} e^{-i\Omega_1(x_1-x_0)} \hat{L}_{s_1,s_0}(\Omega_1)\big(du^{s_0\prime}(x_0) + i\Omega_1 u^{s_0}(x_0)\big) \tilde{\Phi}^{s_1}(x_1)\tilde{\Phi}^{s_0}(x_0).
\end{aligned}
\tag{43}
$$

Now we need to substitute Fourier representation of $u(x)$, and also take into account the symmetry w.r.t. swap of times $x_1$ and $x_2$; in the result,

$$S^{(1)}_{\check{\Phi}} = \frac{g}{\varepsilon_0} \sum_{s_1,s_0} \int dx_1 dx_0 \int \frac{d\Omega_1 d\Omega}{(2\pi)^2} e^{-i\Omega_1(x_1-x_0)} \hat{L}_{s_1,s_0}(\Omega_1)(\Omega d + \Omega_1) e^{-i\Omega x_1} u^{s_1}_\Omega \tilde{\Phi}^{s_1}(x_1)\tilde{\Phi}^{s_0}(x_0). \tag{44}$$

To make expression more symmetric we perform shift $\Omega_1 \to \Omega_1 - \frac{\Omega}{2}$, and then integrate over $x_i$:

$$S^{(1)}_{\check{\Phi}} = \frac{g}{\varepsilon_0} \sum_{s_1,s_0} \int \frac{d\Omega_1 d\Omega}{(2\pi)^2} \hat{L}_{s_1,s_0}\left(\Omega_1 - \frac{\Omega}{2}\right)\left(\Omega(d - \frac{1}{2}) + \Omega_1\right) u^{s_1}_\Omega \tilde{\Phi}^{s_1}_{-\Omega_1-\frac{\Omega}{2}} \tilde{\Phi}^{s_0}_{\Omega_1-\frac{\Omega}{2}}. \tag{45}$$

Finally, we transform (45) into the $(cl,q)$ representation:

$$S^{(1)} = \frac{2g}{\varepsilon_0} \int \frac{d\Omega_1 d\Omega}{(2\pi)^2} \frac{1}{\sqrt{2}} \begin{pmatrix} u^{cl}_\Omega \\ u^{q}_\Omega \end{pmatrix}^T \begin{pmatrix} \hat{\tilde{\Phi}}^T_{-\Omega_1-\frac{\Omega}{2}} L_{I,1}(\Omega,\Omega_1) \hat{\tilde{\Phi}}_{\Omega_1-\frac{\Omega}{2}} \\ \hat{\tilde{\Phi}}^T_{-\Omega_1-\frac{\Omega}{2}} \tau_X L_{I,1}(\Omega,\Omega_1) \hat{\tilde{\Phi}}_{\Omega_1-\frac{\Omega}{2}} \end{pmatrix}, \tag{46}$$

$$L_{I,1}(\Omega,\Omega_1) = \left(\Omega(d - \frac{1}{2}) + \Omega_1\right)\hat{L}\left(\Omega_1 - \frac{\Omega}{2}\right).$$

The last formula concludes our derivation. Here $\tau_X$ is a Pauli matrix $X$ in the space $cl,q$.

### B.4   Calculation of $S^{(2)}$

The second-order terms consist of two groups which will be calculated in sequence.

#### B.4.1   Calculation of $S^{(2,1)}$

After the use of symmetry to the time swap, the expression for $S^{(2,1)}$ reads:

$$S^{(2,1)}_{\check{\Phi}} = \frac{ig}{\varepsilon_0} \sum_{s_1,s_0} \int dx_1 dx_0 \int \frac{d\Omega}{2\pi} e^{-i\Omega(x_1-x_0)} \hat{L}_{s_1,s_0}(\Omega) \times$$

$$\times \left(\frac{d(d-1)}{2}\big([u^{s_1\prime}(x_1)]^2\big) - \frac{\Omega^2}{2}\big([u^{s_1}(x_1)]^2\big) - i\Omega d(u^{s_1}(x_1)u^{s_1\prime}(x_1))\right) \tilde{\Phi}^{s_1}(x_1)\tilde{\Phi}^{s_0}(x_0). \tag{47}$$

We proceed similar to the previous case. After Fourier transformation of $u(x)$ we perform the frequency shift and do the integrals over $x_i$; as a result we obtain:

$$S^{(2,1)}_{\check{\Phi}} = -\frac{ig}{\varepsilon_0} \sum_{s_1,s_0} \int \frac{d\Omega_1 d\Omega_2 d\Omega_2}{(2\pi)^3} \left(\frac{d(d-1)}{2}\Omega_0\Omega_1 + \frac{1}{2}\left(\Omega_2 - \frac{\Omega_1+\Omega_0}{2}\right)^2 + d\left(\Omega_2 - \frac{\Omega_1+\Omega_0}{2}\right)\Omega_1\right) \times$$

$$\times \hat{L}_{s_1,s_0}\left(\Omega_2 - \frac{\Omega_1+\Omega_0}{2}\right) u^{s_1}_{\Omega_0} u^{s_1}_{\Omega_1} \tilde{\Phi}^{s_1}_{-\Omega_2-\frac{\Omega_1+\Omega_0}{2}} \tilde{\Phi}^{s_0}_{\Omega_2-\frac{\Omega_1+\Omega_0}{2}}. \tag{48}$$

Finally we present results in the matrix form:

$$S_{\tilde{\Phi}}^{(2,1)} = -\frac{ig}{\varepsilon_0} \int \frac{d\Omega_1 d\Omega_2 d\Omega_2}{(2\pi)^3} \left( \frac{d(d-1)}{2} \Omega_0 \Omega_1 + \frac{1}{2} \left( \Omega_2 - \frac{\Omega_1 + \Omega_0}{2} \right)^2 + d \left( \Omega_2 - \frac{\Omega_1+\Omega_0}{2} \right) \Omega_1 \right) \times$$

$$\times \hat{u}_{\Omega_0}^T \begin{pmatrix} \hat{\tilde{\Phi}}^T_{-\Omega_2-\frac{\Omega_1+\Omega_0}{2}} \hat{L}\left(\Omega_2-\frac{\Omega_1+\Omega_0}{2}\right) \hat{\tilde{\Phi}}_{\Omega_2-\frac{\Omega_1+\Omega_0}{2}} & \hat{\tilde{\Phi}}^T_{-\Omega_2-\frac{\Omega_1+\Omega_0}{2}} \tau_X \hat{L}\left(\Omega_2-\frac{\Omega_1+\Omega_0}{2}\right) \hat{\tilde{\Phi}}_{\Omega_2-\frac{\Omega_1+\Omega_0}{2}} \\ \hat{\tilde{\Phi}}^T_{-\Omega_2-\frac{\Omega_1+\Omega_0}{2}} \tau_X \hat{L}\left(\Omega_2-\frac{\Omega_1+\Omega_0}{2}\right) \hat{\tilde{\Phi}}_{\Omega_2-\frac{\Omega_1+\Omega_0}{2}} & \hat{\tilde{\Phi}}^T_{-\Omega_2-\frac{\Omega_1+\Omega_0}{2}} \hat{L}\left(\Omega_2-\frac{\Omega_1+\Omega_0}{2}\right) \hat{\tilde{\Phi}}_{\Omega_2-\frac{\Omega_1+\Omega_0}{2}} \end{pmatrix} \hat{u}_{\Omega_1}^T . \quad (49)$$

It concludes our calculation of $S^{(2,1)}$.

### B.4.2  Calculation of $S^{(2,2)}$

This term can be written in the Fourier domain after integration over $x_i$:

$$S_{\tilde{\Phi}}^{(2,2)} = -\frac{ig}{2\varepsilon_0} \sum_{s_1,s_0} \int \frac{d\Omega_2 d\Omega_0 d\Omega_1}{(2\pi)^3} \left( d^2 \Omega_0 \Omega_1 - \left( \Omega_2 - \frac{\Omega_1-\Omega_0}{2} \right)^2 + d \left( \Omega_2 - \frac{\Omega_1-\Omega_0}{2} \right) (\Omega_0 - \Omega_1) \right) \times$$

$$\times u_{\Omega_1}^{s_1} u_{\Omega_0}^{s_0} \hat{L}_{s_1,s_0} \left( \Omega_2 - \frac{\Omega_1-\Omega_0}{2} \right) \tilde{\Phi}_{-\Omega_2-\frac{\Omega_1+\Omega_0}{2}}^{s_1} \tilde{\Phi}_{\Omega_2-\frac{\Omega_1+\Omega_0}{2}}^{s_0} . \quad (50)$$

Writing this expression in $(cl, q)$ notations, we find:

$$S_{\tilde{\Phi}}^{(2,2)} = -\frac{ig}{2\varepsilon_0} \int \frac{d\Omega_2 d\Omega_0 d\Omega_1}{(2\pi)^3} \left( d^2 \Omega_0 \Omega_1 - \left( \Omega_2 - \frac{\Omega_1-\Omega_0}{2} \right)^2 + d \left( \Omega_2 - \frac{\Omega_1-\Omega_0}{2} \right) (\Omega_0 - \Omega_1) \right) \times$$

$$\times \hat{u}_{\Omega_1}^T \begin{pmatrix} \hat{\tilde{\Phi}}^T_{-\Omega_2-\frac{\Omega_1+\Omega_0}{2}} \hat{L}\left(\Omega_2-\frac{\Omega_1-\Omega_0}{2}\right) \hat{\tilde{\Phi}}_{\Omega_2-\frac{\Omega_1+\Omega_0}{2}} & \hat{\tilde{\Phi}}^T_{-\Omega_2-\frac{\Omega_1+\Omega_0}{2}} \hat{L}\left(\Omega_2-\frac{\Omega_1-\Omega_0}{2}\right) \tau_X \hat{\tilde{\Phi}}_{\Omega_2-\frac{\Omega_1+\Omega_0}{2}} \\ \hat{\tilde{\Phi}}^T_{-\Omega_2-\frac{\Omega_1+\Omega_0}{2}} \tau_X \hat{L}\left(\Omega_2-\frac{\Omega_1-\Omega_0}{2}\right) \hat{\tilde{\Phi}}_{\Omega_2-\frac{\Omega_1+\Omega_0}{2}} & \hat{\tilde{\Phi}}^T_{-\Omega_2-\frac{\Omega_1+\Omega_0}{2}} \tau_X \hat{L}\left(\Omega_2-\frac{\Omega_1-\Omega_0}{2}\right) \tau_X \hat{\tilde{\Phi}}_{\Omega_2-\frac{\Omega_1+\Omega_0}{2}} \end{pmatrix} \hat{u}_{\Omega_0} . \quad (51)$$

### B.4.3  Summation of two parts of $S^{(2)}$

Finally, we calculate $S^{(2)} = S^{(2,1)} + S^{(2,2)}$. Both expressions for $S^{(2,1)}$ and $S^{(2,2)}$ share common matrix structure, which contains two $2\times2$ Keldysh spaces, each one for $u(x)$ and $\Phi(x)$ variables. Their sum can represented in the following way in terms of 4 new matrices of rank 2 each:

$$\begin{pmatrix} \hat{L}_{II}^Q(\Omega_0,\Omega_1,\Omega_2) & \hat{L}_{II}^A(\Omega_0,\Omega_1,\Omega_2) \\ \hat{L}_{II}^R(\Omega_0,\Omega_1,\Omega_2) & \hat{L}_{II}^K(\Omega_0,\Omega_1,\Omega_2) \end{pmatrix} \equiv \quad (52)$$

$$\left( d(d-1)\Omega_0\Omega_1 + \left( \Omega_2 - \frac{\Omega_1+\Omega_0}{2} \right)^2 + 2d \left( \Omega_2 - \frac{\Omega_1+\Omega_0}{2} \right) \Omega_0 \right) \begin{pmatrix} \hat{L}\left(\Omega_2-\frac{\Omega_1+\Omega_0}{2}\right) & \tau_X \hat{L}\left(\Omega_2-\frac{\Omega_1+\Omega_0}{2}\right) \\ \tau_X \hat{L}\left(\Omega_2-\frac{\Omega_1+\Omega_0}{2}\right) & \hat{L}\left(\Omega_2-\frac{\Omega_1+\Omega_0}{2}\right) \end{pmatrix} +$$

$$\left( d^2\Omega_0\Omega_1 - \left( \Omega_2 - \frac{\Omega_1-\Omega_0}{2} \right)^2 + d \left( \Omega_2 - \frac{\Omega_1-\Omega_0}{2} \right) (\Omega_0 - \Omega_1) \right) \begin{pmatrix} \hat{L}\left(\Omega_2-\frac{\Omega_1-\Omega_0}{2}\right) & \hat{L}\left(\Omega_2-\frac{\Omega_1-\Omega_0}{2}\right)\tau_X \\ \tau_X \hat{L}\left(\Omega_2-\frac{\Omega_1-\Omega_0}{2}\right) & \tau_X \hat{L}\left(\Omega_2-\frac{\Omega_1-\Omega_0}{2}\right)\tau_X \end{pmatrix} .$$

As a result we can write $S^{(2)}$ as

$$S_{\tilde{\Phi}}^{(2)} = -\frac{ig}{2\varepsilon_0} \int \frac{d\Omega_2 d\Omega_0 d\Omega_1}{(2\pi)^3} \times$$

$$\times \hat{u}_{\Omega_1}^T \begin{pmatrix} \hat{\tilde{\Phi}}^T_{-\Omega_2-\frac{\Omega_1+\Omega_0}{2}} \hat{L}_{II}^Q(\Omega_0,\Omega_1,\Omega_2) \hat{\tilde{\Phi}}_{\Omega_2-\frac{\Omega_1+\Omega_0}{2}} & \hat{\tilde{\Phi}}^T_{-\Omega_2-\frac{\Omega_1+\Omega_0}{2}} \hat{L}_{II}^A(\Omega_0,\Omega_1,\Omega_2) \hat{\tilde{\Phi}}_{\Omega_2-\frac{\Omega_1+\Omega_0}{2}} \\ \hat{\tilde{\Phi}}^T_{-\Omega_2-\frac{\Omega_1+\Omega_0}{2}} \hat{L}_{II}^R(\Omega_0,\Omega_1,\Omega_2) \hat{\tilde{\Phi}}_{\Omega_2-\frac{\Omega_1+\Omega_0}{2}} & \hat{\tilde{\Phi}}^T_{-\Omega_2-\frac{\Omega_1+\Omega_0}{2}} \hat{L}_{II}^K(\Omega_0,\Omega_1,\Omega_2) \hat{\tilde{\Phi}}_{\Omega_2-\frac{\Omega_1+\Omega_0}{2}} \end{pmatrix} \hat{u}_{\Omega_0} . \quad (53)$$

This action leads both the modification of the effective action of soft modes, and to additional contributions to the susceptibility.

## C  Quadratic action of soft modes.

In this section we will show that the combined SYK$_4$ SYK$_2$ model with $\Gamma \gg T$ demonstrates properties which are very different from the pure SYK model. To find the correction to the action of soft mode due to $\Gamma_{ij}$ terms in the Hamiltonian, we set $\hat{\tilde{\Phi}}_\Omega = \Phi_\Omega^{cl}\left(\begin{smallmatrix}1\\0\end{smallmatrix}\right)$. Using this substitution in (53) we can write:

$$S_{\tilde{\Phi}}^{(2)} = -\frac{ig}{2\varepsilon_0}\int\frac{d\Omega_2 d\Omega_0 d\Omega_1}{(2\pi)^3}\tilde{\Phi}_{-\Omega_2-\frac{\Omega_1+\Omega_0}{2}}^{cl}\tilde{\Phi}_{\Omega_2-\frac{\Omega_1+\Omega_0}{2}}^{cl}\hat{u}_{\Omega_1}^T\hat{L}_{II}(\Omega_0,\Omega_1,\Omega_2)\hat{u}_{\Omega_0} \times$$
$$\times\hat{L}_{II}(\Omega_0,\Omega_1,\Omega_2) = \begin{pmatrix}[\hat{L}_{II}^Q(\Omega_0,\Omega_1,\Omega_2)]_{cl,cl} & [\hat{L}_{II}^A(\Omega_0,\Omega_1,\Omega_2)]_{cl,cl}\\ [\hat{L}_{II}^R(\Omega_0,\Omega_1,\Omega_2)]_{cl,cl} & [\hat{L}_{II}^K(\Omega_0,\Omega_1,\Omega_2)]_{cl,cl}\end{pmatrix}$$

(54)

and the consequencies which come due to this additional term in the effective action.

### C.1  Action for the problem without pumping

In the absense of pumping $\tilde{\Phi}_\Omega^{cl} = 2\pi\delta(\Omega)$ and we get the following simple expression for the additional action:

$$\delta S_{soft} = -\frac{ig}{2\varepsilon_0}\int\frac{d\Omega}{2\pi}\hat{u}_{-\Omega}^T\hat{L}_{II}(\Omega,-\Omega,0)\hat{u}_\Omega.$$

(55)

Here it is time to wire $\hat{L}_{II}(\Omega,-\Omega,0)$ explicitly:

$$\hat{L}_{II}(\Omega,-\Omega,0) = -\frac{i\Omega^2}{2}\begin{pmatrix}0 & \psi(\Omega)\\ \psi(-\Omega) & -i\pi\end{pmatrix},\quad \psi(\Omega) = \Psi\left(\frac{1}{2}+i\Omega\right)-\Psi\left(-\frac{1}{2}\right),\quad \Psi(z) = \partial_z\ln\Gamma(z).$$

(56)

The full action quadratic action of the model has form:

$$\delta S_{soft} = \frac{1}{2}\int\frac{d\Omega}{2\pi}\hat{u}_{-\Omega}^T\begin{pmatrix}0 & [\mathcal{G}^A(\Omega)]^{-1}\\ [\mathcal{G}^R(\Omega)]^{-1} & -\mathcal{G}^K(\Omega)[\mathcal{G}^A(\Omega)]^{-1}[\mathcal{G}^R(\Omega)]^{-1}\end{pmatrix}\hat{u}_\Omega,$$
$$[\mathcal{G}^{R(A)}(\Omega)]^{-1} = \Omega^2\left(\varepsilon_0(\Omega^2+1)-\frac{g}{2\varepsilon_0}\psi(\pm\Omega)\right)$$

(57)

$$-\mathcal{G}^K(\Omega)[\mathcal{G}^A(\Omega)]^{-1}[\mathcal{G}^R(\Omega)]^{-1} = i\frac{g}{2\varepsilon_0}\pi.$$

One can makes two observations: first, the distribution function determined by the relation $\mathcal{G}^K(\Omega) \equiv F(\Omega)\left(\mathcal{G}^R(\Omega)-\mathcal{G}^A(\Omega)\right)$ reaches its equilibrium value: $F(\Omega) = \coth(\pi\Omega)$. Secondly, for $g \gg \varepsilon_0^2$, we observe the "resonant" behavior with the resonance frequency $\Omega_R \gg 1$. The position of the resonance and the behavior of the Green function in its vicinity are determined by the relations:

$$\varepsilon_0\left(\Omega_R^2+1\right) = \frac{g}{2\varepsilon_0}\Re\psi(\Omega_R),\quad \mathcal{G}^{R(A)}(\Omega) \approx \frac{1}{2\Omega_R^3\varepsilon_0}\frac{sgn(\Omega)}{\delta\Omega\pm i\Omega_W},$$
$$\Omega = \pm\Omega_R+\delta\Omega,\quad \frac{\Omega_W}{\Omega_R} = -\frac{1}{2}\frac{\Im\psi(\Omega_R)}{\Re\psi(\Omega_R)} \approx \frac{\pi}{4\ln(\Omega_R)}.$$

(58)

### C.2  Action for the problem with pumping

In the problem with pumping we have $\tilde{\Phi}_\Omega = 2\pi\left[\delta(\Omega)+\frac{A}{2}\left(\delta(\Omega-\Omega_P)+\delta(\Omega+\Omega_P)\right)\right]$. In the previous subsection we have described the term without $A$. The term linear in $A$ has the form:

$$\delta^{(1)}S_{soft} = -\frac{ig}{2\varepsilon_0}\frac{A}{2}\int\frac{d\Omega}{(2\pi)}\hat{u}_{-\Omega-\Omega_P}^T L_{P,I,1}(\Omega,\Omega_P)\hat{u}_\Omega + [\Omega_P\to-\Omega_P],$$
$$L_{P,I,1}(\Omega,\Omega_P) = \left[\hat{L}_{II}(\Omega,-\Omega-\Omega_P,\frac{\Omega_P}{2})+\hat{L}_{II}(\Omega,-\Omega-\Omega_P,-\frac{\Omega_P}{2})\right].$$

(59)

We see that slow and fast modes of $u(x)$ fluctuations are coupled due to the pumping term. We integrate now over fast motions to find action for slow mode alone:

$$\delta^{(2,1)}S_{soft} = \frac{i}{2}\langle[\delta^{(1)}S_{soft}]^2\rangle_{fast} =$$

$$\frac{1}{2}\left[\frac{g}{2\varepsilon_0}\frac{A}{2}\right]^2\int\limits_{\Omega\ll\Omega_W}\frac{d\Omega}{(2\pi)}\hat{u}^T_{-\Omega}L_{P,I}(-\Omega,-\Omega_P)^T\mathcal{G}_{\Omega_P}L_{P,I}(\Omega,\Omega_P)\hat{u}_\Omega + [\Omega_P\to-\Omega_P]\,, \quad (60)$$

where

$$L_{P,I}(\Omega,\Omega_P) = \frac{1}{2}\left(L_{P,I,1}(\Omega,\Omega_P) + L_{P,I,1}(-\Omega-\Omega_P,\Omega_P)^T\right). \quad (61)$$

In the main order over $\Omega/\Omega_W\ll 1$ the action in Eq.(60) has the form:

$$\delta^{(2,1)}S_{soft}\approx\pi\Omega_R\frac{ig}{2\varepsilon_0}\left(\frac{A}{2}\right)^2\int\limits_{\Omega\ll\Omega_W}\frac{d\Omega}{(2\pi)}\hat{u}^T_{-\Omega}\begin{pmatrix}0&\Omega\\-\Omega&0\end{pmatrix}\hat{u}_\Omega\,. \quad (62)$$

The term quadratic in $A$ leads to the following correction to the action:

$$\delta^{(2,2)}S_{soft}\approx-\frac{ig}{2\varepsilon_0}\left(\frac{A}{2}\right)^2\int\frac{d\Omega}{(2\pi)}\hat{u}^T_{-\Omega}\left[\hat{L}_{II}(\Omega,-\Omega,-\Omega_P)+(\Omega_P\to-\Omega_P)\right]\hat{u}_\Omega$$

$$\approx\pi\Omega_P\frac{ig}{2\varepsilon_0}\left(\frac{A}{2}\right)^2\int\frac{d\Omega}{(2\pi)}\hat{u}^T_{-\Omega}\begin{pmatrix}0&-2\Omega\\2\Omega&4\Omega_P\end{pmatrix}\hat{u}_\Omega\,. \quad (63)$$

The combination of both contributions gives:

$$\delta_A S_{soft} = \pi\Omega_P\frac{ig}{2\varepsilon_0}\left(\frac{A}{2}\right)^2\int\frac{d\Omega}{(2\pi)}\hat{u}^T_{-\Omega}\begin{pmatrix}0&-\Omega\\\Omega&4\Omega_P\end{pmatrix}\hat{u}_\Omega\,. \quad (64)$$

# D  Calculation of susceptibility

Relative smallness of $u(x)$ fluctuations allows (as explained in the main text) to represent the susceptibility has the following form:

$$\chi(\Omega) = \frac{1}{2}\frac{\delta^2}{\delta f^q_\Omega\delta f^{cl}_\Omega}\left(S^{(0)}_{\bar{\Phi}} + \frac{i}{2}\langle[S^{(1)}]^2\rangle\right). \quad (65)$$

The calculation of $S^{(0)}$ was performed above. It is time to calculate the second term.

## D.1  Calculation of $\frac{i}{2}\langle[S^{(1)}]^2\rangle$

Using the expression for $S^{(1)}$ from Eq.(46) and doing the averaging, we obtain

$$\frac{i}{2}\langle[S^{(1)}]^2\rangle = i\left(\frac{g}{\varepsilon_0}\right)^2\langle\int\frac{d\Omega_1 d\Omega}{(2\pi)^2}\frac{d\Omega_2 d\Omega'}{(2\pi)^2}\begin{pmatrix}\hat{\bar{\Phi}}^T_{-\Omega_1-\frac{\Omega}{2}}L_{I,1}(\Omega,\Omega_1)\hat{\bar{\Phi}}_{\Omega_1-\frac{\Omega}{2}}\\\hat{\bar{\Phi}}^T_{-\Omega_1-\frac{\Omega}{2}}\tau_X L_{I,1}(\Omega,\Omega_1)\hat{\bar{\Phi}}_{\Omega_1-\frac{\Omega}{2}}\end{pmatrix}^T\times \quad (66)$$

$$\times\begin{pmatrix}u^{cl}_\Omega\\u^q_\Omega\end{pmatrix}\begin{pmatrix}u^{cl}_{\Omega'}\\u^q_{\Omega'}\end{pmatrix}^T\begin{pmatrix}\hat{\bar{\Phi}}^T_{-\Omega_2-\frac{\Omega'}{2}}L_{I,1}(\Omega',\Omega_2)\hat{\bar{\Phi}}_{\Omega_2-\frac{\Omega'}{2}}\\\hat{\bar{\Phi}}^T_{-\Omega_2-\frac{\Omega'}{2}}\tau_X L_{I,1}(\Omega',\Omega_2)\hat{\bar{\Phi}}_{\Omega_2-\frac{\Omega'}{2}}\end{pmatrix}\rangle$$

$$=-\left(\frac{g}{\varepsilon_0}\right)^2\int\frac{d\Omega_1 d\Omega d\Omega_2}{(2\pi)^3}\begin{pmatrix}\hat{\bar{\Phi}}^T_{-\Omega_1-\frac{\Omega}{2}}L_{I,1}(\Omega,\Omega_1)\hat{\bar{\Phi}}_{\Omega_1-\frac{\Omega}{2}}\\\hat{\bar{\Phi}}^T_{-\Omega_1-\frac{\Omega}{2}}\tau_X L_{I,1}(\Omega,\Omega_1)\hat{\bar{\Phi}}_{\Omega_1-\frac{\Omega}{2}}\end{pmatrix}^T\hat{\mathcal{G}}_\Omega\begin{pmatrix}\hat{\bar{\Phi}}^T_{-\Omega_2+\frac{\Omega}{2}}L_{I,1}(-\Omega,\Omega_2)\hat{\bar{\Phi}}_{\Omega_2+\frac{\Omega}{2}}\\\hat{\bar{\Phi}}^T_{-\Omega_2+\frac{\Omega}{2}}\tau_X L_{I,1}(-\Omega,\Omega_2)\hat{\bar{\Phi}}_{\Omega_2+\frac{\Omega}{2}}\end{pmatrix}. \quad (67)$$

Here we used the relation $-i\left\langle\begin{pmatrix} u^{cl}_\Omega \\ u^q_\Omega \end{pmatrix}\begin{pmatrix} u^{cl}_{\Omega'} \\ u^q_{\Omega'} \end{pmatrix}^T\right\rangle = 2\pi\delta(\Omega+\Omega')\hat{\mathcal{G}}_\Omega$ where $\hat{\mathcal{G}}_\Omega = \begin{pmatrix} \mathcal{G}^K_\Omega & \mathcal{G}^R_\Omega \\ \mathcal{G}^A_\Omega & 0 \end{pmatrix}$. To calculate susceptibility we need to introduce a probe field by replacing $\tilde{\Phi}$ by $\tilde{\Phi}+f$ after that we should find a second order term in $f$; after that we can use $\tilde{\Phi}$ containing classical component only. We denote the needed quadratic term as $\left\{\frac{i}{2}\langle[S^{(1)}]^2\rangle\right\}^{(2)}_f$ and calculate it now.

## D.2 Calculation of $\left\{\frac{i}{2}\langle[S^{(1)}]^2\rangle\right\}^{(2)}_f$

According to the definition:

$$\left\{\frac{i}{2}\langle[S^{(1)}]^2\rangle\right\}^{(2)}_f = -\left(\frac{g}{\varepsilon_0}\right)^2\int\frac{d\Omega_1 d\Omega d\Omega_2}{(2\pi)^3}\begin{pmatrix}\hat{\tilde{\Phi}}^T_{-\Omega_1-\frac{\Omega}{2}}L_{I,2,1}(\Omega,\Omega_1)\hat{f}_{\Omega_1-\frac{\Omega}{2}} \\ \hat{\tilde{\Phi}}^T_{-\Omega_1-\frac{\Omega}{2}}L_{I,2,2}(\Omega,\Omega_1)\hat{f}_{\Omega_1-\frac{\Omega}{2}}\end{pmatrix}^T\hat{\mathcal{G}}_\Omega\begin{pmatrix}\hat{\tilde{\Phi}}^T_{-\Omega_2+\frac{\Omega}{2}}L_{I,2,1}(-\Omega,\Omega_2)\hat{f}_{\Omega_2+\frac{\Omega}{2}} \\ \hat{\tilde{\Phi}}^T_{-\Omega_2+\frac{\Omega}{2}}L_{I,2,2}(-\Omega,\Omega_2)\hat{f}_{\Omega_2+\frac{\Omega}{2}}\end{pmatrix} \quad (68)$$
$$-2\left(\frac{g}{\varepsilon_0}\right)^2\int\frac{d\Omega_1 d\Omega d\Omega_2}{(2\pi)^3}\begin{pmatrix}\hat{f}^T_{-\Omega_1-\frac{\Omega}{2}}L_{I,2,1}(\Omega,\Omega_1)\hat{f}_{\Omega_1-\frac{\Omega}{2}} \\ \hat{f}^T_{-\Omega_1-\frac{\Omega}{2}}L_{I,2,2}(\Omega,\Omega_1)\hat{f}_{\Omega_1-\frac{\Omega}{2}}\end{pmatrix}^T\hat{\mathcal{G}}_\Omega\begin{pmatrix}\hat{\tilde{\Phi}}^T_{-\Omega_2+\frac{\Omega}{2}}L_{I,1}(-\Omega,\Omega_2)\hat{\tilde{\Phi}}_{\Omega_2+\frac{\Omega}{2}} \\ \hat{\tilde{\Phi}}^T_{-\Omega_2+\frac{\Omega}{2}}\tau_X L_{I,1}(-\Omega,\Omega_2)\hat{\tilde{\Phi}}_{\Omega_2+\frac{\Omega}{2}}\end{pmatrix}.$$

Here $L_{I,2,1}(\Omega,\Omega_1) = L_{I,1}(\Omega,\Omega_1) + L_{I,1}(\Omega,-\Omega_1)^T$ and $L_{I,2,2}(\Omega,\Omega_1) = \tau_X L_{I,1}(\Omega,\Omega_1) + L_{I,1}(\Omega,-\Omega_1)^T\tau_X$. Finally, we need to write this term in more convenient form (also we did not use the classical structure of the field $\tilde{\Phi}$)

$$\left\{\frac{i}{2}\langle[S^{(1)}]^2\rangle\right\}^{(2)}_f = -\left(\frac{g}{\varepsilon_0}\right)^2\int\frac{d\Omega_1 d\Omega d\Omega_2}{(2\pi)^3}\tilde{\Phi}^{cl}_{-\Omega_1-\frac{\Omega}{2}}\tilde{\Phi}^{cl}_{-\Omega_2+\frac{\Omega}{2}}\hat{f}^T_{\Omega_1-\frac{\Omega}{2}}\hat{\Pi}_{I,1}(\Omega,\Omega_1,\Omega_2)\hat{f}_{\Omega_2+\frac{\Omega}{2}}$$
$$-2\left(\frac{g}{\varepsilon_0}\right)^2\int\frac{d\Omega_1 d\Omega d\Omega_2}{(2\pi)^3}\tilde{\Phi}^{cl}_{\Omega_2+\frac{\Omega}{2}}\tilde{\Phi}^{cl}_{-\Omega_2+\frac{\Omega}{2}}\hat{f}^T_{-\Omega_1-\frac{\Omega}{2}}\hat{\Pi}_{I,2}(\Omega,\Omega_1,\Omega_2)\hat{f}_{\Omega_1-\frac{\Omega}{2}},$$
$$\left[\hat{\Pi}_{I,1}(\Omega,\Omega_1,\Omega_2)\right]_{s_1,s_0} = \begin{pmatrix}\left[L_{I,2,1}(\Omega,\Omega_1)\right]_{cl,s_1} \\ \left[L_{I,2,2}(\Omega,\Omega_1)\right]_{cl,s_1}\end{pmatrix}^T\hat{\mathcal{G}}_\Omega\begin{pmatrix}\left[L_{I,2,1}(-\Omega,\Omega_2)\right]_{cl,s_0} \\ \left[L_{I,2,2}(-\Omega,\Omega_2)\right]_{cl,s_0}\end{pmatrix},$$
$$\hat{\Pi}_{I,2}(\Omega,\Omega_1,\Omega_2) = \begin{pmatrix}L_{I,2,1}(\Omega,\Omega_1) \\ L_{I,2,2}(\Omega,\Omega_1)\end{pmatrix}^T\hat{\mathcal{G}}_\Omega\begin{pmatrix}\left[L_{I,1}(-\Omega,\Omega_2)\right]_{cl,cl}, \\ \left[L_{I,1}(-\Omega,\Omega_2)\right]_{q,cl}\end{pmatrix}. \quad (69)$$

Finally, we need to perform shifts of frequencies in the first term to write this result in a simple form:

$$\left\{\frac{i}{2}\langle[S^{(1)}]^2\rangle\right\}^{(2)}_f = -2\left(\frac{g}{\varepsilon_0}\right)^2\int\frac{d\Omega_1 d\Omega d\Omega_2}{(2\pi)^3}\tilde{\Phi}^{cl}_{\Omega_2+\frac{\Omega}{2}}\tilde{\Phi}^{cl}_{-\Omega_2+\frac{\Omega}{2}}\hat{f}^T_{-\Omega_1-\frac{\Omega}{2}}\hat{\Pi}_I(\Omega,\Omega_1,\Omega_2)\hat{f}_{\Omega_1-\frac{\Omega}{2}},$$
$$\hat{\Pi}_I(\Omega,\Omega_1,\Omega_2) = \hat{\Pi}_{I,2}(\Omega,\Omega_1,\Omega_2) + \frac{1}{2}\hat{\Pi}_{I,1}\left(\Omega_1-\Omega_2, -\frac{\Omega_1+\Omega_2+\Omega}{2}, \frac{\Omega_1+\Omega_2-\Omega}{2}\right). \quad (70)$$

### D.3 Susceptibility: major contribution

The major contribution to susceptibility is obtained when we set $\tilde{\Phi}_\Omega = 2\pi\delta(\Omega)$ in the above action (70); pumping then enters via modification of the soft-mode action only. The result for the action then reads:

$$\left\{\frac{i}{2}\langle[S^{(1)}]^2\rangle\right\}^{(2)}_f = -2\left(\frac{g}{\varepsilon_0}\right)^2\int\frac{d\Omega}{(2\pi)}\hat{f}^T_{-\Omega}\hat{\Pi}_I(0,\Omega,0)\hat{f}_\Omega,$$
$$\hat{\Pi}_I(0,\Omega,0) = \frac{\Omega^2}{2}\begin{pmatrix}0 & \mathcal{G}^A_\Omega\psi^2(\Omega) \\ \mathcal{G}^R_\Omega\psi^2(-\Omega) & -i\pi\left[\mathcal{G}^A_\Omega\psi(\Omega)+\mathcal{G}^R_\Omega\psi(-\Omega)\right]+\mathcal{G}^K_\Omega\psi(-\Omega)\psi(\Omega)\end{pmatrix}, \quad (71)$$

and susceptibility has now the following from:

$$\boxed{\chi(\Omega) = -\frac{2g}{\varepsilon_0}\psi(-\Omega)\left[1 + \frac{g}{2\varepsilon_0}\Omega^2\mathcal{G}^R_\Omega\psi(-\Omega)\right].} \quad (72)$$

Here we have omitted the constant (frequency-independent) term related with the UV logarithmic divergence at the scale $\frac{J}{T}$. We emphasize that Green function in Eq.(72) cointains, in general, the effect of pumping. In the explicit form the susceptibility is given by

$$\chi(\Omega) = -\frac{2g}{\varepsilon_0}\psi(-\Omega)\frac{(\Omega^2+1)}{(\Omega^2+1)-\frac{g}{2\varepsilon_0^2}\psi(-\Omega)}, \tag{73}$$

$$\Im\chi^R(\Omega) = -\frac{2g}{\varepsilon_0}\Im\psi(-\Omega)\frac{(\Omega^2+1)^2}{\left(\Omega^2+1-\frac{g}{2\varepsilon_0^2}\Re\psi(-\Omega)\right)^2+\left[\frac{g}{2\varepsilon_0^2}\Im\psi(-\Omega)\right]^2}. \tag{74}$$

### D.4 Susceptibility: additional term

In the presence of the pumping we can write: $\tilde{\Phi}_\Omega = 2\pi\left(\delta(\Omega) + \frac{A}{2}\left(\delta(\Omega-\Omega_P) + \Omega_P \to -\Omega_P\right)\right)$. Taking into account the terms $\sim A$ in $\tilde{\Phi}_\Omega$ results in the additional term of the order $A^2$ in the action

$$\delta_A\left\{\frac{i}{2}\langle[S^{(1)}]^2\rangle\right\}_f^{(2)} = -2\left(\frac{Ag}{2\varepsilon_0}\right)^2\int\frac{d\Omega}{(2\pi)}\hat{f}_{-\Omega}^T\left[\hat{\Pi}_I(0,\Omega,\Omega_P) + \Omega_P \to -\Omega_P\right]\hat{f}_\Omega$$

$$\approx i\frac{\ln(\Omega_R)A^2}{\Omega_W}\frac{g}{\varepsilon_0}\int\frac{d\Omega}{(2\pi)}\hat{f}_{-\Omega}^T\begin{pmatrix}0 & -\Omega \\ \Omega & \Omega_P\end{pmatrix}\hat{f}_\Omega, \tag{75}$$

which leads to the following contribution to the susceptibility: $\delta_A\chi^R(\Omega) = i\frac{\ln(\Omega_R)A^2}{\Omega_W}\frac{g}{\varepsilon_0}\Omega$. It is smaller than the major contribution as $\Omega/\Omega_W \ll 1$.

## E Heating

Pumping leads to absorption of energy and thus to heating of the system. For the SYK model the change of temperature leads to the non-zero average of $\partial_x\langle u^{cl}(x)\frac{1}{\sqrt{2}}\rangle = \frac{T_{new}}{T_{old}}$ so for sufficiently-long heating we will break our assumption $\partial_x u \ll 1$. On the other hand, as it was shown in the main text, dissipation rate for "dry friction" regime does not depend on the temperature; thus our result is not sensitive to heating, as long as we neglect the terms in the action beyond quadratic approximation over soft mode $u(x)$.

To keep the increase of absorbed energy and heating $\delta T/T$ small, we consider pumping with small amplitude $A$ and finite duration $t_{\text{pump}}$, which is non-monochromatic by definition. In this Section we will show how to relate spectrum of our pulse and the change of $u^{cl}$. We will consider the general case of the pumping in the form $\tilde{\Phi}_\Omega^{cl} = (2\pi\delta(\Omega) + \Phi_\Omega)$ and $\Phi^q = 0$. Our aim to write an expression for $\langle u_\Omega^{cl}\rangle$, but first we need to find the propagator of fluctuations of the soft mode. In presence of pumping it is no longer a function of a single frequency, since the problem is non-stationary.

We assume that pumping is weak and use perturbation theory up to the second order terms in the pumping amplitude. In the absence of the pumping we have: $\mathcal{G}^{(0)}(\Omega,\Omega') = 2\pi\delta(\Omega+\Omega')\mathcal{G}_\Omega$. where $\mathcal{G}_\Omega$ is determined in the subsection III.A. The linear correction to the Green function can be obtained from the expression for $S_{\tilde{\Phi}}^{(2)}$ and has the form:

$$\mathcal{G}^{(1)}(\Omega,\Omega') = \frac{ig}{2\varepsilon_0}\Phi_{\Omega+\Omega'}\mathcal{G}_\Omega\left[\hat{L}_{II}(-\Omega',-\Omega,\frac{\Omega+\Omega'}{2}) + \hat{L}_{II}(-\Omega',-\Omega,-\frac{\Omega+\Omega'}{2})\right]\mathcal{G}_{-\Omega'}. \tag{76}$$

Using Eq.(46) from appendix B.3 for $S^{(1)}$ we find for the average values:

$$\left\langle\begin{pmatrix}u_\Omega^{(cl)} \\ u_\Omega^{(q)}\end{pmatrix}\right\rangle = -\frac{2g}{\varepsilon_0}\int\frac{d\Omega_1 d\Omega'}{(2\pi)^2}\frac{\tilde{\Phi}_{-\Omega_1-\frac{\Omega'}{2}}^{cl}\tilde{\Phi}_{\Omega_1-\frac{\Omega'}{2}}^{cl}}{\sqrt{2}}\mathcal{G}(\Omega,\Omega')\begin{pmatrix}0 \\ 1\end{pmatrix}L_{I,1}(\Omega',\Omega_1)_{q,cl}. \tag{77}$$

We are interesting in the term proportional to the second power of the pumping amplitude $A$. Since $S^{(1)} \propto A$, it is sufficient to use in Eq.(77) the first-order correction $\mathcal{G}^{(1)}$ to the Green function. Our aim is to calculate

$$\lim_{\Omega \to 0} -\Omega^2 \frac{1}{\sqrt{2}} \langle u_\Omega \rangle = \partial_x \frac{1}{\sqrt{2}} \langle u(x) \rangle |_{-\infty}^{\infty} = \frac{\delta T}{T}. \tag{78}$$

Using Eq.(77) we find:

$$\frac{\delta T}{T} = -\lim_{\Omega \to 0} \langle \frac{\Omega^2}{\sqrt{2}} u_\Omega^{(cl)} \rangle = \left[ \lim_{\Omega \to 0} \Omega^2 G(\Omega) \right] \int \frac{d\Omega'}{2\pi} |\Phi_{\Omega'}|^2 \mathfrak{I} \chi(\Omega') \frac{\Omega'}{2}. \tag{79}$$

In terms of physical frequency $\omega$ the result reads

$$\delta T = \frac{1}{2\pi} \frac{1}{\varepsilon_0 + \frac{g}{\varepsilon_0}} \int \frac{d\omega'}{2\pi} |\Phi_{\omega'}|^2 \mathfrak{I} \chi(\omega') \frac{\omega'}{2} = \frac{Q}{C}, \tag{80}$$

where

$$Q = \int \frac{d\omega'}{2\pi} |\Phi_{\omega'}|^2 \mathfrak{I} \chi(\omega') \frac{\omega'}{2} \tag{81}$$

is the full energy absorption and

$$C = 2\pi \left( \varepsilon_0 + \frac{g}{\varepsilon_0} \right) \tag{82}$$

is the heat capacity, compare with SM, the end of appendix B.2 and Eq.(42). For a pumping pulse with frequency $\omega_R$, amplitude $A$ and duration $t_{\text{pump}}$, we have (with $\delta = 1/t_{\text{pump}}$):

$$\Phi_\omega = \frac{A}{2} \left[ \frac{\delta}{\delta^2 + (\omega - \omega_R)^2} + \frac{\delta}{\delta^2 + (\omega + \omega_R)^2} \right], \tag{83}$$

therefore

$$Q = \frac{A^2}{16} \omega_R \, t_{\text{pump}} \mathfrak{I} \chi(\omega_R). \tag{84}$$

The results (84,82) were used to derive inequality (24) of the main text.

## F   Modification of the saddle-point solution

In the major part of this paper we have studied the properties of the soft reparametrization mode and its impact upon the susceptibility, in presence of quadratic terms $\propto \Gamma_{ij}$. Below we consider different effect of these terms, that is, modification of the saddle-point solution for the fermionic Green function; in other terms, here we account for the "hard modes" effect.

To analyze hard modes we use the following action:

$$S = S_\varphi - \frac{i}{2} N \sum_{s,s'} \int dx dx' \left[ \mathcal{M} s s' G_{ss'}^2(x, x') + G_{s,s'}(x, x') \Sigma_{s,s'}(x, x') \right]$$

$$- i \frac{N}{2} Tr \ln \left( \hat{1} + \hat{G}^\varphi \circ \hat{\Sigma} \right), \tag{85}$$

where $\mathcal{M} \equiv \frac{\Gamma^2}{2J^2} \left( \frac{2\pi}{\beta J} \right)^{-1}$. Eq.(85) contains soft-mode dependent Green function

$$G_{s_1, s_2}^\varphi(x_1, x_2) = G_{s_1, s_2}^0(\varphi_{s_1}(x_1), \varphi_{s_2}(x_2)) \left[ \varphi_{s_1}'(x_1) \varphi_{s_2}'(x_2) \right]^\Delta, \tag{86}$$

where $\hat{G}^0$ is the conformal solution for the $SYK$ model, which has the following form in the $(cl, q)$ basis, in the Fourier domain:

$$\hat{G} = \begin{pmatrix} \mathcal{G}^R(\Omega) & \mathcal{G}^K(\Omega) \\ 0 & \mathcal{G}^A(\Omega) \end{pmatrix} = -2ib^{\Delta}\cos(\pi\Delta)\begin{pmatrix} K_{\Delta}(\Omega) & \tanh(\pi\Omega)\{K_{\Delta}(\Omega) + K_{\Delta}(-\Omega)\} \\ 0 & -K_{\Delta}(-\Omega) \end{pmatrix}. \quad (87)$$

In the presence of the source field the action has the form:

$$S = S_{\varphi} - \frac{i}{2}N\sum_{s,s'}\int dx\,dx' \times \Big[\mathcal{M}(1 + \Phi^s(x))\big(1 + \Phi^{s'}(x')\big)ss'G^2_{ss'}(x,x')$$
$$+ G_{s,s'}(x,x')\Sigma_{s,s'}(x,x')\Big] - i\frac{N}{2}Tr\ln\big(\hat{1} + \hat{G}^{\varphi}\circ\hat{\Sigma}\big). \quad (88)$$

It is useful to rewrite it using matrix notations:

$$S = S_{\varphi} + \frac{i}{2}N\mathcal{M}Tr\big[\{\hat{1} + \hat{\mathcal{F}}\}\circ G\circ\{\hat{1} + \hat{\mathcal{F}}\}\circ G\big] + \frac{i}{2}NTr\big[\hat{G}\circ\hat{\Sigma}\big] - i\frac{N}{2}Tr\ln\big(\hat{1} + \hat{G}^{\varphi}\circ\hat{\Sigma}\big),$$
$$\hat{\mathcal{F}}(x,x') = \begin{pmatrix} \Phi^{cl}(x) & \Phi^q(x) \\ \Phi^q(x) & \Phi^{cl}(x) \end{pmatrix}\delta(x - x'). \quad (89)$$

In the limit $\mathcal{M} \ll 1$ fluctuations near the saddle point of the field $G$ and $\Sigma$ are small. Let us consider the effect of these fluctuations assuming that $G = G^{\varphi} + \delta G$ and $\Sigma = \delta\Sigma$. In this case:

$$S = S_{\varphi} + \frac{i}{2}N\mathcal{M}Tr\big[\{\hat{1} + \hat{\mathcal{F}}\}\circ\hat{G}^{\varphi}\circ\{\hat{1} + \hat{\mathcal{F}}\}\circ\hat{G}^{\varphi}\big] +$$
$$iN\mathcal{M}Tr\big[\{\hat{1} + \hat{\mathcal{F}}\}\circ\delta\hat{G}\circ\{\hat{1} + \hat{\mathcal{F}}\}\circ(\hat{G}^{\varphi})\big] + \frac{i}{2}N\mathcal{M}Tr\big[\{\hat{1} + \hat{\mathcal{F}}\}\circ\delta\hat{G}\circ\{\hat{1} + \hat{\mathcal{F}}\}\circ\delta\hat{G}\big]$$
$$+ \frac{i}{2}NTr\big[(\hat{G}^{\varphi} + \delta\hat{G})\circ\hat{\Sigma}\big] - i\frac{N}{2}Tr\big(\hat{G}^{\varphi}\circ\delta\hat{\Sigma} - \frac{1}{2}\hat{G}^{\varphi}\circ\delta\hat{\Sigma}\circ\hat{G}^{\varphi}\circ\delta\hat{\Sigma}\big). \quad (90)$$

The first line is the action of our original model (soft-mode only) and we had calculated the corresponding susceptibility already. The second term in the second line is unimportant since 1) this correction to the action is small as $\mathcal{M} \ll 1$, and 2) it contains two $\delta G$ fields and thus the correction to susceptibility will contain extra small factor $1/N \ll 1$. We also note that in the leading order there is no mixing between soft mode $\varphi$ and fields $G$ and $\Sigma$. As a result the correction to susceptibility is determined by the action:

$$\delta S\big[\tfrac{i}{2}N\big]^{-1} = 2\mathcal{M}Tr\big[\delta\hat{G}\circ\{\hat{1} + \hat{\mathcal{F}}\}\circ(\hat{G}^0)\circ\{\hat{1} + \hat{\mathcal{F}}\}\big] + Tr\big[\delta\hat{G}\circ\hat{\Sigma}\big] \quad (91)$$
$$+ \frac{1}{2}Tr\big(\hat{G}^{\varphi}\circ\hat{\Sigma}\circ\hat{G}^{\varphi}\circ\hat{\Sigma}\big).$$

After integration over $\delta G$ and $\delta\Sigma$ we have the following action :

$$S_{\mathcal{F}}\Big[\frac{i}{2}N\Big]^{-1} = 2(2\mathcal{M})^2\,Tr\big[\hat{\mathcal{F}}\circ\hat{G}^0\circ\hat{\mathcal{F}}\circ\hat{G}^0\circ\hat{G}^0\circ\hat{G}^0\big] +$$
$$(2\mathcal{M})^2\,Tr\big[\hat{\mathcal{F}}\circ\hat{G}^0\circ\hat{G}^0\circ\hat{\mathcal{F}}\circ\hat{G}^0\circ\hat{G}^0\big]. \quad (92)$$

The action (92) can be used to calculate correction to the susceptibility; the result is

$$\delta\chi(\Omega) = -i4N\mathcal{M}^2\int\frac{d\Omega'}{2\pi}F_{\Omega'}\big(\{G^R_{\Omega'} - G^A_{\Omega'}\}\big\{[G^R_{\Omega'+\Omega}]^3 + [G^A_{\Omega'-\Omega}]^3\big\}$$
$$+ \big\{[G^R_{\Omega'}]^3 - [G^A_{\Omega'}]^3\big\}\{G^R_{\Omega'+\Omega} + G^A_{\Omega'-\Omega}\} + \big\{[G^R_{\Omega'}]^2 - [G^A_{\Omega'}]^2\big\}\big\{[G^R_{\Omega'+\Omega}]^2 + [G^A_{\Omega'-\Omega}]^2\big\}\big). \quad (93)$$

Taking the imaginary part of Eq.(93) in the low-$\Omega$ limit, we obtain Eq.(23) of the main text.

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
