# Peer review of "Non-equilibrium Sachdev-Ye-Kitaev model with quadratic perturbation"

_SciPost Physics, doi:SciPost Phys. 12, 031 (2022)_

## Round 2 · Referee Report · Anonymous (Referee 1) · 2021-10-18

Report

The paper studies the SYK model with a quadratic perturbation and builds upon results by the same authors in previous works. The main result is a computation of the absorption rate in a model with a periodically modulated quadratic fermion term. The authors find that even though the quadratic perturbation does not significantly change the saddle, the absorption rate is significantly different from pure SYK. The results are interesting and I recommend it for publication.

Some comments:

In eqs. (7), (8), (14) I did not understand why the quadratic term $S^{(2)}$ is included with fluctuations $S_{SYK}$, while $e^{S^{(1)}}$ is expanded out. Naively, it seems that $S^{(2)}$ should be suppressed in comparison to $S^{(1)}$ and could also be treated as a small perturbation. Then the computation could be done with the SYK propagator defined by $S_{SYK}$.
Minor: a) There are some inconsistencies in the text when referring to the “action eq. 5”.Presumably the unnumbered equation between (4) and (5) is implied. b) Integration variable in eq. 30 seems to be misprinted.

---

## Round 2 · Referee Report · Anonymous (Referee 2) · 2021-11-8

Report

It is a very interesting paper which further deepens the understanding of the so-called reparametrization modes in the non-equilibrium setting of the SYK model perturbed by quadratic Majorana couplings. In particular, authours are able to identify the very clear manifestation of the polaron type of physics which was previously discovered by them using the gravity dual description.

The results of the paper seems to be reasonable and well theoretically justified. So, I will definitely recommend this manuscript for the publication in SciPost after some revision.

My main concern is about the clarity of presentation. It would be nice if authors elucidate a physical meaning of their starting action (7). For instance, drawing few standard diagrams showing an external pumping force and the reparametrization mode joined into 3 and 4 point vertices would help a lot for potential readers to easily grasp the physics. The same diagrams could be later used to show different contributions to the pumping rate.

One could also perhaps add the citation to

Skvortsov et al., JETP Lett. 80, 54 (2004), "Energy absorption in time-dependent unitary random matrix ensembles: dynamic vs Anderson localization"

As to me, the present manuscript constitutes a natural extension of the same problem to the case of interacting fermions.

---

## Round 3 · Referee Report · Anonymous · 2021-11-11

Report

The authors addressed my questions. I recommend publication.

---

## Round 3 · Referee Report · Anonymous · 2021-11-26

Report

The authors have included my suggestions in somewhat different way that I have expected. However, I agree that the resubmitted version has a better readability as compared to the previous one. Comments of the 1st Referee have also lead to the improvement of an overall presentation. Thus, I suggest that the manuscript can be now published in its present form.

---

## Round 3 · Author Response

Resubmission latter.

We would like to thank the referees for their reports and comments.
The following detailed answers to their comments follow below.

To the report #1:
Minor comments: a&b – we have done all the corrections proposed
Main comments: We have retained S_2 terms as it changes the quadratic action of the soft modes as this contribution is dominant for low frequencies. It can be seen from Eq.10 where the Green function of the soft modes is presented. The second term in the brackets (with function $\psi$) comes from the S_2 and is dominant in the assumption $\Gamma \gg T$. We do not need to take into account higher terms as they are small as $N\gg1$. The term $S_2$ is also important as it mixes low and high frequencies in the non-linear regime (see eq. (28) ).
To the report #2:
In our model, interaction (described by the SYK model) is much stronger than quadratic perturbation. As a result, the starting point of perturbative analysis is an SYK model, which does not have a standard diagrammatic description with an ordinary fermionic Green function. Moreover, the fluctuation propagator in the SYK model has a “zero mode” due to asymptotic symmetry. This mode could be taken into account using the path integral approach for the variable $\phi$ (as action is non-local we could not write a Hamiltonian). As a result, for the SYK model, the diagrammatic description could be used in terms of fluctuation propagator of $\phi$. To clarify the origin of the key relation shown in Eq.(18), we added the text between equations (18)-(22) which shows how susceptibility can be obtained working in the functional integral representation for the $\phi$ field.
We also added the citation of the mentioned paper by Skvortsov et al. It is referred to in the Introduction after the words: “In other words, we propose here a generalization of the approach well-known [6–9]”

Aleksey Lunkin and Mikhail Feigel’man

---

## Round 3 · List of Changes

1. The citation of the paper by Skvortsov et al. was added in the introduction.
2. The equation (5) was enumerated
3. Intervals in the equation (8) were increased
4. We add comments between equations (18) and (22).
5. The last line in the equation (35) was corrected.

---

## Editorial Decision

published